# Genetic correlation between amyotrophic lateral sclerosis and schizophrenia

Russell L. McLaughlin[1,2,*], Dick Schijven[3,4,*], Wouter van Rheenen[3], Kristel R. van Eijk[3], Margaret O'Brien[1], Project MinE GWAS Consortium[†], Schizophrenia Working Group of the Psychiatric Genomics Consortium[‡], René S. Kahn[4], Roel A. Ophoff[4,5,6], An Goris[7], Daniel G. Bradley[2], Ammar Al-Chalabi[8], Leonard H. van den Berg[3], Jurjen J. Luykx[3,4,9,**], Orla Hardiman[2,**] & Jan H. Veldink[3,**]

We have previously shown higher-than-expected rates of schizophrenia in relatives of patients with amyotrophic lateral sclerosis (ALS), suggesting an aetiological relationship between the diseases. Here, we investigate the genetic relationship between ALS and schizophrenia using genome-wide association study data from over 100,000 unique individuals. Using linkage disequilibrium score regression, we estimate the genetic correlation between ALS and schizophrenia to be 14.3% (7.05–21.6; $P = 1 \times 10^{-4}$) with schizophrenia polygenic risk scores explaining up to 0.12% of the variance in ALS ($P = 8.4 \times 10^{-7}$). A modest increase in comorbidity of ALS and schizophrenia is expected given these findings (odds ratio 1.08–1.26) but this would require very large studies to observe epidemiologically. We identify five potential novel ALS-associated loci using conditional false discovery rate analysis. It is likely that shared neurobiological mechanisms between these two disorders will engender novel hypotheses in future preclinical and clinical studies.

[1] Academic Unit of Neurology, Trinity Biomedical Sciences Institute, Trinity College Dublin, Dublin DO2 DK07, Republic of Ireland. [2] Smurfit Institute of Genetics, Trinity College Dublin, Dublin D02 DK07, Republic of Ireland. [3] Department of Neurology and Neurosurgery, Brain Center Rudolf Magnus, University Medical Center Utrecht, Utrecht 3584 CX, The Netherlands. [4] Department of Psychiatry, Brain Center Rudolf Magnus, University Medical Center Utrecht, Utrecht 3584 CX, The Netherlands. [5] Department of Human Genetics, David Geffen School of Medicine, University of California, Los Angeles, California 90095, USA. [6] Center for Neurobehavioral Genetics, Semel Institute for Neuroscience and Human Behavior, University of California, Los Angeles, California 90095, USA. [7] Department of Neurosciences, Experimental Neurology and Leuven Research Institute for Neuroscience and Disease (LIND), KU Leuven—University of Leuven, Leuven B-3000, Belgium. [8] Department of Basic and Clinical Neuroscience, Maurice Wohl Clinical Neuroscience Institute, King's College London, London WC2R 2LS, UK. [9] Department of Psychiatry, Hospital Network Antwerp (ZNA) Stuivenberg and Sint Erasmus, Antwerp 2020, Belgium. * These authors contributed equally to this work. ** These authors jointly supervised this work. Correspondence and requests for materials should be addressed to R.L.M. (email: mclaugr@tcd.ie).
† A full list of Project MinE GWAS Consortium members appears at the end of the paper. ‡ A full list of Schizophrenia Working Group of the Psychiatric Genomics Consortium members appears at the end of the paper.

A myotrophic lateral sclerosis (ALS) is a late-onset neurodegenerative condition characterized by progressive loss of upper and lower motor neurons, leading to death from respiratory failure in 70% of patients within 3 years of symptom onset. Although ALS is often described as a primarily motor-system disease, extramotor involvement occurs in up to 50% of cases, with prominent executive and behavioural impairment, and behavioural variant frontotemporal dementia (FTD) in up to 14% of cases[1]. A neuropsychiatric prodrome has been described in some people with ALS–FTD, and higher rates of schizophrenia and suicide have been reported in first and second degree relatives of those with ALS, particularly in kindreds associated with the *C9orf72* hexanucleotide repeat expansion[2]. These clinical and epidemiological observations suggest that ALS and schizophrenia may share heritability.

ALS and schizophrenia both have high heritability estimates (0.65 and 0.64, respectively)[3,4]; however the underlying genetic architectures of these heritable components appear to differ. Analysis of large genome-wide association study (GWAS) datasets has implicated over 100 independent risk loci for schizophrenia[5] and estimated that a substantial proportion (23%) of the variance in underlying liability for schizophrenia is due to additive polygenic risk (many risk-increasing alleles of low individual effect combining to cause disease) conferred by common genetic variants[6]. This proportion, the single nucleotide polymorphism (SNP)-based heritability, is lower in ALS (8.2%), in which fewer than ten risk loci have been identified by GWAS[7]. Nevertheless, both diseases have polygenic components, but the extent to which they overlap has not been investigated.

Recently, methods to investigate overlap between polygenic traits using GWAS data have been developed[8–10]. These methods assess either pleiotropy (identical genetic variants influencing both traits) or genetic correlation (identical alleles influencing both traits). Genetic correlation is related to heritability; for both measures, binary traits such as ALS and schizophrenia are typically modelled as extremes of an underlying continuous scale of liability to develop the trait. If two binary traits are genetically correlated, their liabilities covary, and this covariance is determined by both traits having identical risk alleles at overlapping risk loci. Studies of pleiotropy and genetic correlation have provided insights into the overlapping genetics of numerous traits and disorders, although none to date has implicated shared polygenic risk between neurodegenerative and neuropsychiatric disease. Here, we apply several techniques to identify and dissect the polygenic overlap between ALS and schizophrenia. We provide evidence for genetic correlation between the two disorders which is unlikely to be driven by diagnostic misclassification and we demonstrate a lack of polygenic overlap between ALS and other neuropsychiatric and neurological conditions, which could be due to limited power given the smaller cohort sizes for these studies.

## Results

### Genetic correlation between ALS and schizophrenia.
To investigate the polygenic overlap between ALS and schizophrenia, we used individual-level and summary data from GWAS for ALS[7] (36,052 individuals) and schizophrenia[5] (79,845 individuals). At least 5,582 control individuals were common to both datasets, but for some cohorts included in the schizophrenia dataset this could not be ascertained so this number is likely to be higher. For ALS, we used summary data from both mixed linear model association testing[11] and meta-analysis of cohort-level logistic regression[12]. We first used linkage disequilibrium (LD) score regression with ALS and schizophrenia summary statistics; this technique models, for polygenic traits, a linear relationship between a SNP's LD score (the amount of genetic variation that it captures) and its

GWAS test statistic[13]. This distinguishes confounding from polygenicity in GWAS inflation and the regression coefficient can be used to estimate the SNP-based heritability ($h_S^2$) for single traits[13]. In the bivariate case, the regression coefficient estimates genetic covariance ($\rho_g$) for pairs of traits, from which genetic correlation ($r_g$) is estimated[8]; these estimates are unaffected by sample overlap between traits. Using constrained intercept LD score regression with mixed linear model ALS summary statistics, we estimated the liability-scale SNP-based heritability of ALS to be 8.2% (95% confidence interval = 7.2–9.1; mean $\chi^2 = 1.13$; all ranges reported below indicate 95% confidence intervals), replicating previous estimates based on alternative methods[7]. Estimates based on ALS meta-analysis summary statistics and free-intercept LD score regression with mixed linear model summary statistics were lower (Supplementary Table 1), resulting in higher genetic correlation estimates (Supplementary Table 2); for this reason, we conservatively use constrained intercept genetic correlation estimates for ALS mixed linear model summary statistics throughout the remainder of this paper. Heritability estimates for permuted ALS data were null (Supplementary Table 1).

LD score regression estimated the genetic correlation between ALS and schizophrenia to be 14.3% (7.05–21.6; $P = 1 \times 10^{-4}$). Results were similar for a smaller schizophrenia cohort of European ancestry (21,856 individuals)[14], indicating that the inclusion of individuals of Asian ancestry in the schizophrenia cohort did not bias this result (Supplementary Fig. 1). In addition to schizophrenia, we estimated genetic correlation with ALS using GWAS summary statistics for bipolar disorder[15], major depressive disorder[16], attention deficit-hyperactivity disorder[17], autism spectrum disorder[17], Alzheimer's disease (Supplementary Note 1)[18], multiple sclerosis[19] and adult height[20], finding no significant genetic correlation between ALS and any secondary trait other than schizophrenia (Fig. 1; Supplementary Table 2).

**Polygenic risk score analysis.** We supported the positive genetic correlation between ALS and schizophrenia by analysis of

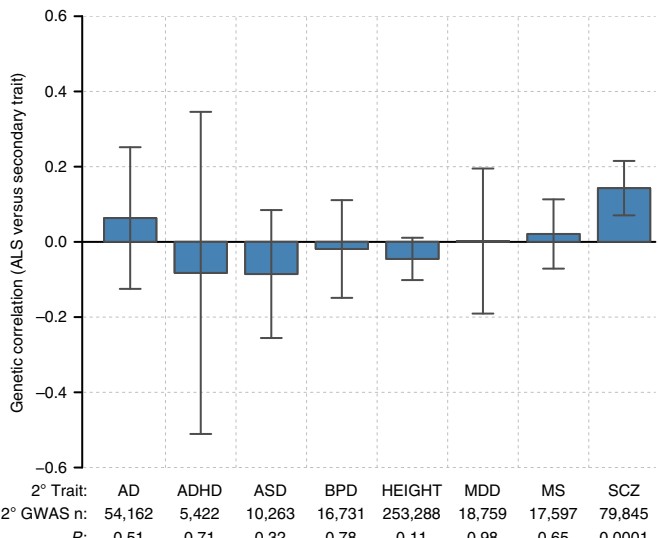

| 2° Trait: | AD | ADHD | ASD | BPD | HEIGHT | MDD | MS | SCZ |
|---|---|---|---|---|---|---|---|---|
| 2° GWAS n: | 54,162 | 5,422 | 10,263 | 16,731 | 253,288 | 18,759 | 17,597 | 79,845 |
| P: | 0.51 | 0.71 | 0.32 | 0.78 | 0.11 | 0.98 | 0.65 | 0.0001 |

**Figure 1 | Genetic correlation between ALS and eight secondary traits.** Error bars indicating 95% confidence intervals and *P*-values were calculated by the LD score regression software using a block jackknife procedure. Secondary traits are: AD, Alzheimer's disease; ADHD, attention deficit-hyperactivity disorder; ASD, autism spectrum disorder; BPD, bipolar disorder; MDD, major depressive disorder; MS, multiple sclerosis; SCZ, schizophrenia.

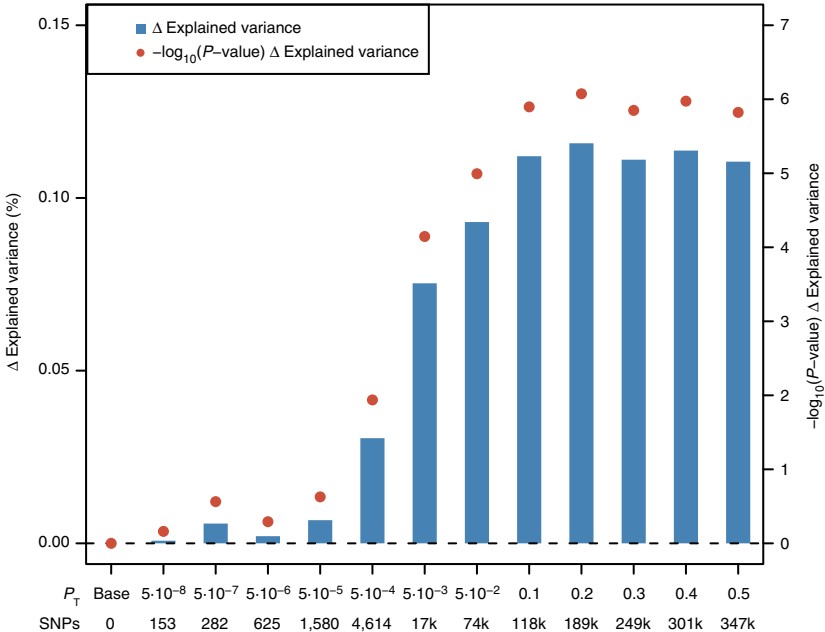

**Figure 2 | Analysis of PRS for schizophrenia in a target sample of 10,032 ALS cases and 16,627 healthy controls.** $P$-value thresholds ($P_T$) for schizophrenia SNPs are shown on the $x$ axis, where the number of SNPs increases with a more lenient $P_T$. Δ Explained variances (Nagelkerke $R^2$, shown as a %) of a generalized linear model including schizophrenia-based PRS versus a baseline model without polygenic scores (blue bars) are shown for each $P_T$. $-\mathrm{Log}_{10}$ $P$-values of Δ explained variance per $P_T$ (red dots) represent $P$-values from the binomial logistic regression of ALS phenotype on PRS, accounting for LD (Supplementary Table 4) and including sex and significant principal components as covariates (Supplementary Fig. 2). Values are provided in Supplementary Table 5.

polygenic risk for schizophrenia in the ALS cohort. Polygenic risk scores (PRS) are per-individual scores based on the sum of alleles associated with one phenotype, weighted by their effect size, measured in an independent target sample of the same or a different phenotype[10]. PRS calculated on schizophrenia GWAS summary statistics for twelve $P$-value thresholds ($P_T$) explained up to 0.12% ($P_T = 0.2$, $P = 8.4 \times 10^{-7}$) of the phenotypic variance in a subset of the individual-level ALS genotype data that had all individuals removed that were known or suspected to be present in the schizophrenia cohort (Fig. 2; Supplementary Table 5). ALS cases had on average higher PRS for schizophrenia compared to healthy controls and harbouring a high schizophrenia PRS for $P_T = 0.2$ significantly increased the odds of being an ALS patient in our cohort (Fig. 3; Supplementary Table 6). Permutation of case–control labels reduced the explained variance to values near zero (Supplementary Fig. 3).

**Modelling misdiagnosis and comorbidity.** Using BUHMBOX[21], a tool that distinguishes true genetic relationships between diseases (pleiotropy) from spurious relationships resulting from heterogeneous mixing of disease cohorts, we determined that misdiagnosed cases in the schizophrenia cohort (for example, young-onset FTD–ALS) did not drive the genetic correlation estimate between ALS and schizophrenia ($P = 0.94$). Assuming a true genetic correlation of 0%, we estimated the required rate of misdiagnosis of ALS as schizophrenia to be 4.86% (2.47–7.13) to obtain the genetic correlation estimate of 14.3% (7.05–21.6; Supplementary Table 7), which we consider to be too high to be likely. However, if ALS and schizophrenia are genetically correlated, more comorbidity would be expected than if the genetic correlation was 0%. Modelling our observed genetic correlation of 14.3% (7.05–21.6), we estimated the odds ratio for having above-threshold liability for ALS given above-threshold liability for schizophrenia to be 1.17 (1.08–1.26), and the same for schizophrenia given ALS (Supplementary Fig. 4). From a clinical

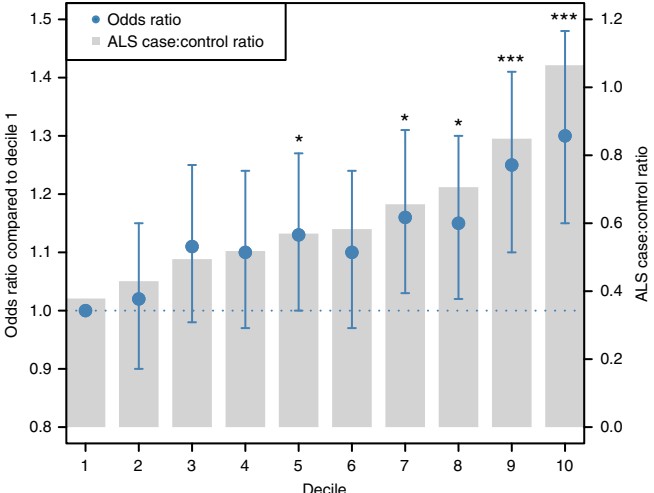

**Figure 3 | Odds ratio for ALS by PRS deciles for schizophrenia.** The figure applies to schizophrenia $P$-value threshold ($P_T$) = 0.2. The PRS for this threshold were converted to ten deciles containing near identical numbers of individuals. Decile 1 contained the lowest scores and decile 10 contained the highest scores, where decile 1 was the reference and deciles 2–10 were dummy variables to contrast to decile 1 for OR calculation. The case:control ratio per decile is indicated with grey bars. Error bars indicate 95% confidence intervals. Significant differences from decile 1 were determined by logistic regression of ALS phenotype on PRS decile, including sex and principal components as covariates and are indicated by *$P < 0.05$ or ***$P < 0.001$.

perspective, to achieve 80% power to detect a significant ($\alpha = 0.05$) excess of schizophrenia in the ALS cohort as a result of this genetic correlation, the required population-based incident cohort size is 16,448 ALS patients (7,310–66,670).

**Pleiotropic risk loci**. We leveraged the genetic correlation between ALS and schizophrenia to discover novel ALS-associated genomic loci by conditional false discovery rate (cFDR) analysis[9,22] (Fig. 4; Supplementary Table 8). Five loci already known to be involved in ALS were identified (corresponding to *MOBP*, *C9orf72*, *TBK1*, *SARM1* and *UNC13A*) along with five potential novel loci at cFDR < 0.01 (*CNTN6*, *TNIP1*, *PPP2R2D*, *NCKAP5L* and *ZNF295-AS1*). No gene set was significantly enriched (after Bonferroni correction) in genome-wide cFDR values when analysed using MAGENTA.

## Discussion

There is evolving clinical, epidemiological and biological evidence for an association between ALS and psychotic illness, particularly schizophrenia. Genetic evidence of overlap to date has been based primarily on individual genes showing Mendelian inheritance, in particular the *C9orf72* hexanucleotide repeat expansion, which is associated with ALS and FTD, and with psychosis in relatives of ALS patients[2]. In this study, we have replicated SNP-based heritability estimates for ALS and schizophrenia using GWAS summary statistics, and have for the first time demonstrated significant overlap between the polygenic components of both diseases, estimating the genetic correlation to be 14.3%. We have carefully controlled for confounding bias, including population stratification and shared control samples, and have shown through analysis of polygenic risk scores that the overlapping polygenic risk applies to SNPs that are modestly associated with both diseases. Given that our genetic correlation estimate relates to the polygenic components of ALS ($h_S^2 = 8.2\%$) and schizophrenia ($h_S^2 = 23\%$) and these estimates do not represent all heritability for both diseases, the accuracy of using schizophrenia-based PRS to predict ALS status in any patient is expected to be low (Nagelkerke's $R^2 = 0.12\%$ for $P_T = 0.2$), although statistically significant ($P = 8.4 \times 10^{-7}$). Nevertheless, the positive genetic correlation of 14.3% indicates that the direction of effect of risk-increasing and protective alleles is consistently aligned between ALS and schizophrenia, suggesting convergent biological mechanisms between the two diseases.

Although phenotypically heterogeneous, both ALS and schizophrenia are clinically recognizable as syndromes[23,24]. The common biological mechanisms underlying the association between the two conditions are not well understood, but are likely associated with disruption of cortical networks. Schizophrenia is a polygenic neurodevelopmental disorder characterized by a combination of positive symptoms (hallucinations and delusions), negative symptoms (diminished motivation, blunted affect, reduction in spontaneous speech and poor social functioning) and impairment over a broad range of cognitive abilities[25]. ALS is a late onset complex genetic disease characterized by a predominantly motor phenotype with recently recognized extra-motor features in 50% of patients, including cognitive impairment[1]. It has been suggested that the functional effects of risk genes in schizophrenia converge by modulating synaptic plasticity, and influencing the development and stabilization of cortical microcircuitry[5]. In this context, our identification of *CNTN6* (contactin 6, also known as NB-3, a neural adhesion protein important in axon development)[26] as a novel pleiotropy-informed ALS-associated locus supports neural network dysregulation as a potential convergent mechanism of disease in ALS and schizophrenia.

No significantly enriched biological pathway or ontological term was identified within genome-wide cFDR values using MAGENTA. Low inflation in ALS GWAS statistics, coupled with a rare variant genetic architecture[7], render enrichment-based biological pathway analyses with current sample sizes challenging. Nevertheless, nine further loci were associated with ALS risk at cFDR < 0.01. Of these, *MOBP*, *C9orf72*, *TBK1*, *SARM1* and *UNC13A* have been described previously in ALS and were associated by cFDR analysis in this study owing to their strong association with ALS through GWAS[7]. The remaining four loci (*TNIP1*, *PPP2R2D*, *NCKAP5L* and *ZNF295-AS1*) are novel associations and may represent pleiotropic disease loci. *TNIP1* encodes TNFAIP3 interacting protein 1 and is involved in autoimmunity and tissue homoeostasis[27]. The protein product of *PPP2R2D* is a regulatory subunit of protein phosphatase 2 and has a role in PI3K-Akt signalling and mitosis[28]. *NCKAP5L* is a homologue of *NCKAP5*, encoding NAP5, a proline-rich protein that has previously been implicated in schizophrenia, bipolar disorder and autism[29,30]. *ZNF295-AS1* is a noncoding RNA[31]. Further investigation into the biological roles of these genes may yield novel insight into the pathophysiology of certain subtypes of ALS and schizophrenia, and as whole-genome and exome datasets become available in the future for appropriately large ALS case–control cohorts, testing for burden of rare genetic variation across these genes will be particularly instructive, especially given the role that rare variants appear to play in the pathophysiology of ALS[7].

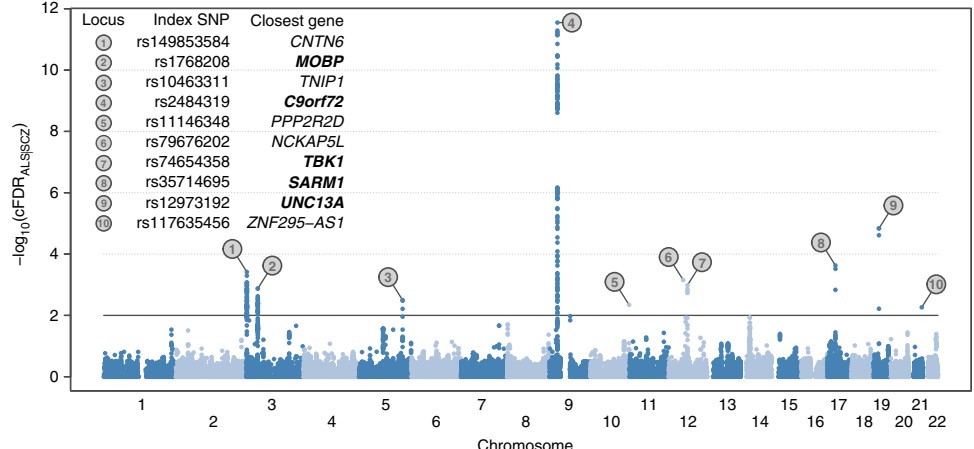

**Figure 4 | Pleiotropy-informed ALS risk loci determined by analysis of cFDR in ALS GWAS *P*-values given schizophrenia GWAS *P*-values (cFDR$_{ALS|SCZ}$).** Each point denotes a SNP; its *x* axis position corresponds to its chromosomal location and its height indicates the extent of association with ALS by cFDR analysis. The solid line indicates the threshold cFDR = 0.01. Any gene whose role in ALS is already established is in bold. A complete list of all loci at cFDR ≤ 0.05 is provided in Supplementary Table 8.

                                                                                     

Our data suggest that other neuropsychiatric conditions (bipolar disorder, autism and major depression) do not share polygenic risk with ALS. This finding contrasts with our recent observations from family aggregation studies and may be unexpected given the extensive genetic correlation between neuropsychiatric conditions[6]. This could relate to statistical power conferred by secondary phenotype cohort sizes, and future studies with larger sample sizes will shed further light on the relationship between ALS and neuropsychiatric disease. It is also possible that the current study underestimates genetic correlations due to the substantial role that rare variants play in the genetic architecture of ALS[7] and future fine-grained studies examining heritability and genetic correlation in low-minor allele frequency and low-LD regions may identify a broader relationship between ALS and neuropsychiatric diseases.

A potential criticism of this study is that the polygenic overlap between ALS and schizophrenia could be driven by misdiagnosis, particularly in cases of ALS–FTD, which can present in later life as a psychotic illness and could be misdiagnosed as schizophrenia. This is unlikely, as strict diagnostic criteria are required for inclusion of samples in the schizophrenia GWAS dataset[5]. Furthermore, since core schizophrenia symptoms are usually diagnosed during late adolescence, a misdiagnosis of FTD-onset ALS–FTD as schizophrenia is unlikely. In this study, we found no evidence for misdiagnosis of ALS as schizophrenia (BUHMBOX $P = 0.94$) and we estimated that a misdiagnosis of 4.86% of ALS cases would be required to spuriously observe a genetic correlation of 14.3%, which is not likely to occur in clinical practice. We are therefore confident that this genetic correlation estimate reflects a genuine polygenic overlap between the two diseases and is not a feature of cohort ascertainment, but the possibility of some misdiagnosis in either cohort cannot be entirely excluded based on available data.

A positive genetic correlation between ALS and schizophrenia predicts an excess of patients presenting with both diseases. Most neurologists and psychiatrists, however, will not readily acknowledge that these conditions co-occur frequently. Our genetic correlation estimate confers an odds ratio of 1.17 (1.08–1.26) for harbouring above-threshold liability for ALS given schizophrenia (or vice versa) and a lifetime risk of 1:34,336 for both phenotypes together. Thus, a very large incident cohort of 16,448 ALS patients (7,310–66,670), with detailed phenotype information, would be required to have sufficient power to detect an excess of schizophrenia within an ALS cohort. Coupled with reduced life expectancy in patients with schizophrenia[32], this may explain the relative dearth of epidemiological studies to date providing clinical evidence of excess comorbidity. Moreover, it has also been proposed that prolonged use of antipsychotic medication may protect against developing all of the clinical features of ALS[33], which would reduce the rate of observed comorbidity. Considering our novel evidence for a genetic relationship between ALS and schizophrenia, this underscores the intriguing possibility that therapeutic strategies for each condition may be useful in the other, and our findings provide rationale to consider the biology of ALS and schizophrenia as related in future drug development studies. Indeed, the glutamate-modulating ALS therapy riluzole has shown efficacy as an adjunct to risperidone, an antipsychotic medication, in reducing the negative symptoms of schizophrenia[34].

In conclusion, we have estimated the genetic correlation between ALS and schizophrenia to be 14.3% (7.05–21.6), providing molecular genetic support for our epidemiological observation of psychiatric endophenotypes within ALS kindreds. To our knowledge, this is the first study to show genetic correlation derived from polygenic overlap between neuro-degenerative and neuropsychiatric phenotypes. The presence of both apparent monogenic C9orf72-driven overlap[2] and polygenic overlap in the aetiology of ALS and schizophrenia suggests the presence of common biological processes, which may relate to disruption of cortical circuitry. As both ALS and schizophrenia are heterogeneous conditions, further genomic, biological and clinical studies are likely to yield novel insights into the pathological processes for both diseases and will provide clinical sub-stratification parameters that could drive novel drug development for both neurodegenerative and psychiatric conditions.

## Methods

**Study population and genetic data.** For ALS, 7,740,343 SNPs genotyped in 12,577 ALS patients and 23,475 healthy controls of European ancestry organized in 27 platform- and country-defined strata were used[7]. The schizophrenia dataset comprised GWAS summary statistics for 9,444,230 SNPs originally genotyped in 34,241 patients and 45,604 controls of European and Asian ancestry[5]. For LD score regression, GWAS summary statistics were generated for the ALS cohort using mixed linear model association testing implemented in Genome-wide Complex Trait Analysis[11] or logistic regression combined with cross-stratum meta-analysis using METAL[12]. To evaluate sample overlap for PRS and cFDR analyses, we also obtained individual-level genotype data for 27,647 schizophrenia cases and 33,675 controls from the schizophrenia GWAS (Psychiatric Genomics Consortium[5] and dbGaP accession number phs000021.v3.p2). Using 88,971 LD-pruned (window size 200 SNPs; shift 20 SNPs; $r^2 > 0.25$) SNPs in both datasets (INFO score > 0.8; MAF > 0.2), with SNPs in high-LD regions removed (Supplementary Table 4), samples were removed from the ALS dataset if they were duplicated or had a cryptically related counterpart (PLINK $\hat{\pi} > 0.1$; 5,582 individuals) in the schizophrenia cohort and whole strata (representing Finnish and German samples; 3,811 individuals) were also removed if commonality with the schizophrenia cohort could not be ascertained (due to unavailability of individual-level genotype data in the schizophrenia cohort) and in which a sample overlap was suspected (Supplementary Table 3).

**LD score regression.** We calculated LD scores using LDSC v1.0.0 in 1 centiMorgan windows around 13,307,412 non-singleton variants genotyped in 379 European individuals (CEU, FIN, GBR, IBS and TSI populations) in the phase 1 integrated release of the 1,000 Genomes Project[35]. For regression weights[13], we restricted LD score calculation to SNPs included in both the GWAS summary statistics and HapMap phase 3; for $r_g$ estimation in pairs of traits this was the intersection of SNPs for both traits and HapMap. Because population structure and confounding were highly controlled in the ALS summary statistics by the use of mixed linear model association tests, we constrained the LD score regression intercept to 1 for $h_S^2$ estimation in ALS, and we also estimated $h_S^2$ with a free intercept. For $h_S^2$ estimation in all other traits and for $r_g$ estimation the intercept was a free parameter. We also estimated $r_g$ using ALS meta-analysis results[7] with free and constrained intercepts and with permuted data conserving population structure. Briefly, principal component analysis was carried out for each stratum using smartpca[36] and the three-dimensional space defined by principal components 1–3 was equally subdivided into 1,000 cubes. Within each cube, case–control labels were randomly swapped and association statistics were re-calculated for the entire stratum using logistic regression. Study-level $P$-values were then calculated using inverse variance weighted fixed effect meta-analysis implemented in METAL[7,12]. $h_S^2$ was estimated for these meta-analysed permuted data using LD score regression (Supplementary Table 1).

**Polygenic risk score analysis.** We calculated PRS for 10,032 cases and 16,627 healthy controls in the ALS dataset (duplicate and suspected or confirmed related samples with the schizophrenia dataset removed), based on schizophrenia-associated alleles and effect sizes reported in the GWAS summary statistics for 6,843,674 SNPs included in both studies and in the phase 1 integrated release of the 1,000 Genomes Project[35] (imputation INFO score < 0.3; minor allele frequency < 0.01; A/T and G/C SNPs removed). SNPs were clumped in two rounds (physical distance threshold of 250 kb and a LD threshold ($R^2$) of > 0.5 in the first round and a distance of 5,000 kb and LD threshold of > 0.2 in the second round) using PLINK v1.90b3y, removing high-LD regions (Supplementary Table 4), resulting in a final set of 496,548 SNPs for PRS calculations. Odds ratios for autosomal SNPs reported in the schizophrenia summary statistics were log-converted to beta values and PRS were calculated using PLINK's score function for twelve schizophrenia GWAS $P$-value thresholds ($P_T$): $5 \times 10^{-8}$, $5 \times 10^{-7}$, $5 \times 10^{-6}$, $5 \times 10^{-5}$, $5 \times 10^{-4}$, $5 \times 10^{-3}$, 0.05, 0.1, 0.2, 0.3, 0.4 and 0.5. A total of 100 principal components (PCs) were generated for the ALS sample using GCTA version 1.24.4. Using R version 3.2.2, a generalized linear model was applied to model the phenotype of individuals in the ALS dataset. PCs that had a significant effect on the phenotype ($P < 0.0005$, Bonferroni-corrected for 100 PCs) were selected (PCs 1, 4, 5, 7, 8, 10, 11, 12, 14, 36, 49).

To estimate explained variance of PRS on the phenotype, a baseline linear relationship including only sex and significant PCs as variables was modelled first:

$$y = \alpha + \beta_{sex} x_{sex} + \sum_n \beta_{pc_n} x_{pc_n},$$

where $y$ is the phenotype in the ALS dataset, $\alpha$ is the intercept of the model with a slope $\beta$ for each variable $x$.

Subsequently, a linear model including polygenic scores for each schizophrenia $P_T$ was calculated:

$$y = \alpha + \beta_{sex} x_{sex} + \sum_n \beta_{pc_n} x_{pc_n} + \beta_{prs} x_{prs}.$$

A Nagelkerke $R^2$ value was obtained for every model and the baseline Nagelkerke $R^2$ value was subtracted, resulting in a $\Delta$ explained variance that describes the contribution of schizophrenia-based PRS to the phenotype in the ALS dataset. PRS analysis was also performed in permuted case–control data (1,000 permutations, conserving case–control ratio) to assess whether the increased $\Delta$ explained variance was a true signal associated with phenotype. $\Delta$ explained variances and $P$-values were averaged across permutation analyses.

To ensure we did not over- or under-correct for population effects in our model, we tested the inclusion of up to a total of 30 PCs in the model, starting with the PC with the most significant effect on the ALS phenotype (Supplementary Fig. 2). Increasing the number of PCs initially had a large effect on the $\Delta$ explained variance, but this effect levelled out after 11 PCs. On the basis of this test we are confident that adding the 11 PCs that had a significant effect on the phenotype sufficiently accounted for possible confounding due to population differences.

For the schizophrenia $P_T$ for which we obtained the highest $\Delta$ explained variance (0.2), we subdivided observed schizophrenia-based PRS in the ALS cohort into deciles and calculated the odds ratio for being an ALS case in each decile compared to the first decile using a similar generalized linear model:

$$y = \alpha + \beta_{sex} x_{sex} + \sum_n \beta_{pc_n} x_{pc_n} + \beta_{decile} x_{decile}.$$

Odds ratios and 95% confidence intervals for ALS were derived by calculating the exponential function of the beta estimate of the model for each of the deciles 2–10.

**Diagnostic misclassification.** To distinguish the contribution of misdiagnosis from true genetic pleiotropy we used BUHMBOX[21] with 417 independent ALS risk alleles in a sample of 27,647 schizophrenia patients for which individual-level genotype data were available. We also estimated the required misdiagnosis rate $M$ of FTD–ALS as schizophrenia that would lead to the observed genetic correlation estimate as $C/(C + 1)$, where $C = \rho_g N_{SCZ}/N_{ALS}$ and $N_{SCZ}$ and $N_{ALS}$ are the number of cases in the schizophrenia and ALS datasets, respectively[37] (derived in Supplementary Methods 1).

**Expected comorbidity.** To investigate the expected comorbidity of ALS and schizophrenia given the observed genetic correlation, we modelled the distribution in liability for ALS and schizophrenia as a bivariate normal distribution with the liability-scale covariance determined by LD score regression (Supplementary Methods 2). Lifetime risks for ALS[38] and schizophrenia[25] of 1/400 and 1/100, respectively, were used to calculate liability thresholds above which individuals develop ALS or schizophrenia, or both. The expected proportions of individuals above these thresholds were used to calculate the odds ratio of developing ALS given schizophrenia, or vice versa (Supplementary Methods 2). The required population size to observe a significant excess of comorbidity was calculated using the binomial power equation.

**Pleiotropy-informed risk loci for ALS.** Using an adapted cFDR method[9] that allows shared controls between cohorts[22], we estimated per-SNP cFDR given LD score-corrected[8] schizophrenia GWAS $P$-values for ALS mixed linear model summary statistics calculated in a dataset excluding Finnish and German cohorts (in which suspected control overlap could not be determined), but including all other overlapping samples (totalling 5,582). To correct for the relationship between LD and GWAS test statistics, schizophrenia summary statistics were residualized on LD score by subtracting the product of each SNP's LD score and the univariate LD score regression coefficient for schizophrenia. cFDR values conditioned on these residualized schizophrenia GWAS $P$-values were calculated for mixed linear model association statistics at 6,843,670 SNPs genotyped in 10,147 ALS cases and 22,094 controls. Pleiotropic genomic loci were considered statistically significant if cFDR < 0.01 (following Andreassen et al.[9]) and were clumped with all neighbouring SNPs based on LD ($r^2 > 0.1$) in the complete ALS dataset. Associated cFDR genomic regions were then mapped to the locations of known RefSeq transcripts in human genome build GRCh37. Genome-wide cFDR values were also tested for enrichment in 9,711 gene sets included in the MAGENTA software package (version 2.4, July 2011) and derived from databases such as Gene Ontology (GO, http://geneontology.org/), Kyoto Encyclopedia of Genes and Genomes (KEGG, http://www.kegg.jp/), Protein ANalysis THrough Evolutionary Relationships (PANTHER, http://www.pantherdb.org/) and INGENUITY (http://www.ingenuity.com/). SNPs were mapped to genes including 20 kb up- and downstream regions to include regulatory elements. The enrichment cutoff applied in our analysis was based on the 95th percentile of gene scores for all genes in the genome. The null distribution of gene scores for each gene set was based on 10,000 randomly sampled gene sets with equal size. MAGENTA uses a Mann–Whitney rank-sum test to assess gene-set enrichment[39].

**Data availability.** All data used in this study are publically available and can be accessed via the studies cited in the text. Other data are available from the authors upon reasonable request.

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

## Acknowledgements

We acknowledge helpful contributions from Mr Gert Jan van de Vendel in the design and execution of PRS analyses. This study received support from the ALS Association; Fondation Thierry Latran; the Motor Neurone Disease Association of England, Wales and Northern Ireland; Science Foundation Ireland; Health Research Board (Ireland), The Netherlands ALS Foundation (Project MinE, to J.H.V., L.H.v.d.B.), the Netherlands Organisation for Health Research and Development (Vici scheme, L.H.v.d.B.) and ZonMW under the frame of E-Rare-2, the ERA Net for Research on Rare Diseases (PYRAMID). Research leading to these results has received funding from the European Community's Health Seventh Framework Programme (FP7/2007–2013). A.G. is supported by the Research Foundation KU Leuven (C24/16/045). A.A.-C. received salary support from the National Institute for Health Research (NIHR) Dementia Biomedical Research Unit and Biomedical Research Centre in Mental Health at South London and Maudsley NHS Foundation Trust and King's College London. The views expressed are those of the authors and not necessarily those of the NHS, the NIHR or the Department of Health. Samples used in this research were in part obtained from the UK National DNA Bank for MND Research, funded by the MND Association and the Wellcome Trust. We acknowledge sample management undertaken by Biobanking Solutions funded by the Medical Research Council (MRC) at the Centre for Integrated Genomic Medical Research, University of Manchester. This is an EU Joint Programme-Neuro-degenerative Disease Research (JPND) Project (STRENGTH, SOPHIA). In addition to those mentioned above, the project is supported through the following funding organizations under the aegis of JPND: UK, Economic and Social Research Council; Italy, Ministry of Health and Ministry of Education, University and Research; France, L'Agence nationale pour la recherche. The work leading up to this publication was funded by the European Community's Health Seventh Framework Programme (FP7/2007–2013; Grant Agreement Number 2,59,867). We thank the International Genomics of Alzheimer's Project (IGAP) for providing summary results data for these analyses. The investigators within IGAP provided data but did not participate in analysis or writing of this report. IGAP was made possible by the generous participation of the control subjects, the patients, and their families. The i-Select chips was funded by the French National Foundation on Alzheimer's disease and related disorders. EADI was supported by the LABEX (laboratory of excellence program investment for the future) DISTALZ grant, Inserm, Institut Pasteur de Lille, Université de Lille 2 and the Lille University Hospital. GERAD was supported by the MRC (Grant No. 5,03,480), Alzheimer's Research UK (Grant No. 5,03,176), the Wellcome Trust (Grant No. 082604/2/07/Z) and German Federal Ministry of Education and Research: Competence Network Dementia Grant no. 01GI0102, 01GI0711, 01GI0420. CHARGE was partly supported by the NIH/NIA Grant R01 AG033193 and the NIA AG081220 and AGES contract N01-AG-12,100, the NHLBI Grant R01 HL105756, the Icelandic Heart Association, and the Erasmus Medical Center and Erasmus University. ADGC was supported by the NIH/NIA Grants: U01 AG032984, U24 AG021886, U01 AG016976, and the Alzheimer's Association Grant ADGC-10-196728. The Project MinE GWAS Consortium included contributions from the PARALS registry, SLALOM group, SLAP registry, FALS Sequencing Consortium, SLAGEN Consortium and NNIPPS Study Group; the Schizophrenia Working Group of the Psychiatric Genomics Consortium included contributions from the Psychosis Endophenotypes International Consortium and Wellcome Trust Case–control Consortium. Members of these eight consortia are listed in Supplementary Note 2.

## Author contributions

O.H., J.H.V. and A.A.-C. conceived the study. R.L.McL., D.S., W.v.R., K.R.v.E., M.O'B., D.G.B., A.A.-C., L.H.v.d.B., J.J.L., O.H. and J.H.V. contributed to study design. R.L.McL., D.S. and W.v.R. conducted the analyses. R.L.McL., D.S., O.H., J.J.L. and J.H.V. drafted the manuscript. R.S.K., R.A.O. and A.G. provided data and critical revision of the manuscript. The Project MinE GWAS Consortium and Schizophrenia Working Group of the Psychiatric Genomics Consortium provided data. R.L.Mc.L. and D.S. contributed equally. J.J.L., O.H. and J.H.V. jointly directed the work.

## Additional information

**Competing interests:** O.H. has received speaking honoraria from Novartis, Biogen Idec, Sanofi Aventis and Merck-Serono. She has been a member of advisory panels for Biogen Idec, Allergen, Ono Pharmaceuticals, Novartis, Cytokinetics and Sanofi Aventis. She serves as Editor-in-Chief of Amyotrophic Lateral Sclerosis and Frontotemporal Dementia. L.H.v.d.B. serves on scientific advisory boards for Prinses Beatrix Spierfonds, Thierry Latran Foundation, Baxalta, Cytokinetics and Biogen, serves on the Editorial Board of the Journal of Neurology, Neurosurgery, and Psychiatry, Amyotrophic Lateral Sclerosis and Frontotemporal Degeneration, and Journal of Neuromuscular Diseases. A.A.-C. has served on advisory panels for Biogen Idec, Cytokinetics, GSK, OrionPharma and Mitsubishi-Tanabe, serves on the Editorial Boards of Amyotrophic Lateral Sclerosis and Frontotemporal Degeneration and F1000, and receives royalties for The Brain: A Beginner's Guide, OneWorld Publications, and Genetics of Complex Human Diseases, Cold Spring Harbor Laboratory Press. The remaining authors declare no competing financial interests.

**How to cite this article**: McLaughlin, R. L. *et al.* Genetic correlation between amyotrophic lateral sclerosis and schizophrenia. *Nat. Commun.* **8,** 14774 doi: 10.1038/ncomms14774 (2017).

## Project MinE GWAS Consortium

Aleksey Shatunov[8], Annelot M. Dekker[3], Frank P. Diekstra[3], Sara L. Pulit[4], Rick A.A. van der Spek[3], Perry T.C. van Doormaal[3], William Sproviero[8], Ashley R. Jones[8], Garth A. Nicholson[10,11], Dominic B. Rowe[10], Roger Pamphlett[12], Matthew C. Kiernan[13], Denis Bauer[14], Tim Kahlke[14], Kelly Williams[10], Filip Eftimov[15], Isabella Fogh[8,16], Nicola Ticozzi[16,17], Kuang Lin[8], Stéphanie Millecamps[18], François Salachas[19], Vincent Meininger[19], Mamede de Carvalho[20,21], Susana Pinto[20,21], Jesus S. Mora[22], Ricardo Rojas-García[23], Meraida Polak[24], Siddharthan Chandran[25,26], Shuna Colville[25], Robert Swingler[25], Karen E. Morrison[27,28], Pamela J. Shaw[29], John Hardy[30], Richard W. Orrell[31], Alan Pittman[30,32], Katie Sidle[31], Pietro Fratta[33], Andrea Malaspina[34,35], Susanne Petri[36], Susanna Abdulla[37], Carsten Drepper[38], Michael Sendtner[38], Thomas Meyer[39], Martina Wiedau-Pazos[5], Catherine Lomen-Hoerth[40], Vivianna M. Van Deerlin[41], John Q. Trojanowski[41], Lauren Elman[42], Leo McCluskey[42], Nazli Basak[43], Thomas Meitinger[44], Peter Lichtner[44], Milena Blagojevic-Radivojkov[44], Christian R. Andres[45], Cindy Maurel[45], Gilbert Bensimon[46], Bernhard Landwehrmeyer[47], Alexis Brice[48], Christine A.M. Payan[46], Safa Saker-Delye[49], Alexandra Dürr[50], Nicholas Wood[51], Lukas Tittmann[52], Wolfgang Lieb[52], Andre Franke[53], Marcella Rietschel[54], Sven Cichon[55,56,57,58,59], Markus M. Nöthen[55,56], Philippe Amouyel[60], Christophe Tzourio[61], Jean-François Dartigues[61], Andre G. Uitterlinden[62,63], Fernando Rivadeneira[62,63], Karol Estrada[62], Albert Hofman[63], Charles Curtis[64], Anneke J. van der Kooi[15], Marianne de Visser[15], Markus Weber[65], Christopher E. Shaw[8], Bradley N. Smith[8], Orietta Pansarasa[66], Cristina Cereda[66], Roberto Del Bo[67], Giacomo P. Comi[67], Sandra D'Alfonso[68], Cinzia Bertolin[69], Gianni Sorarù[69], Letizia Mazzini[70], Viviana Pensato[71], Cinzia Gellera[71], Cinzia Tiloca[16], Antonia Ratti[16,17], Andrea Calvo[72,73], Cristina Moglia[72,73], Maura Brunetti[72,73], Simon Arcuti[74], Rosa Capozzo[74], Chiara Zecca[74], Christian Lunetta[75], Silvana Penco[76], Nilo Riva[77], Alessandro Padovani[78], Massimiliano Filosto[78], Ian Blair[10], P. Nigel Leigh[79], Federico Casale[72], Adriano Chio[72,73], Ettore Beghi[80], Elisabetta Pupillo[80], Rosanna Tortelli[74], Giancarlo Logroscino[81,82], John Powell[8], Albert C. Ludolph[47], Jochen H. Weishaupt[47], Wim Robberecht[83], Philip Van Damme[83,84], Robert H. Brown[85], Jonathan Glass[24], John E. Landers[85], Peter M. Andersen[46,86], Philippe Corcia[87,88], Patrick Vourc'h[45], Vincenzo Silani[16,17], Michael A. van Es[3], R. Jeroen Pasterkamp[89], Cathryn M. Lewis[90,91] & Gerome Breen[6,92,93]

[10]Department of Biomedical Sciences, Faculty of Medicine and Health Sciences, Macquarie University, Sydney, New South Wales, Australia. [11]University of Sydney, ANZAC Research Institute, Concord Hospital, Sydney, New South Wales, Australia. [12]The Stacey MND Laboratory, Department of Pathology, The University of Sydney, Sydney, New South Wales, Australia. [13]Brain and Mind Research Institute, The University of, Sydney, New South Wales, Australia. [14]Transformational Bioinformatics, Commonwealth Scientific and Industrial Research Organisation, Sydney, New South Wales, Australia. [15]Department of Neurology, Academic Medical Center, Amsterdam, The Netherlands. [16]Department of Neurology and Laboratory of Neuroscience, IRCCS Istituto Auxologico Italiano, Milano, Italy. [17]Department of Pathophysiology and Tranplantation, 'Dino Ferrari' Center, Università degli Studi di Milano, Milano, Italy. [18]Institut du Cerveau et de la Moelle épinière, Inserm U1127, CNRS UMR 7225, Sorbonne Universités, UPMC Univ Paris 06 UMRS1127, Paris, France. [19]Ramsay Generale de Santé, Hopital Peupliers, Centre SLA Ile de France, Paris, France. [20]Institute of Physiology and Institute of Molecular Medicine, University of Lisbon, Lisbon, Portugal. [21]Department of Neurosciences, Hospital de Santa Maria-CHLN, Lisbon, Portugal. [22]Department of Neurology, Hospital Carlos III, Madrid, Spain. [23]Neurology Department, Hospital de la Santa Creu i Sant Pau de Barcelona, Autonomous University of Barcelona, Barcelona, Spain. [24]Department Neurology and Emory ALS Center, Emory University School of Medicine, Atlanta, Georgia, USA. [25]Euan MacDonald Centre for Motor Neurone Disease Research, Edinburgh, UK. [26]Centre for Neuroregeneration and Medical Research Council Centre for Regenerative Medicine, University of Edinburgh, Edinburgh, UK. [27]School of Clinical and Experimental Medicine, College of Medical and Dental Sciences, University of Birmingham, Birmingham, UK. [28]Queen Elizabeth Hospital, University Hospitals Birmingham NHS Foundation Trust, Birmingham, UK. [29]Sheffield Institute for Translational Neuroscience (SITraN), University of Sheffield, Sheffield, UK. [30]Department of Molecular Neuroscience, Institute of Neurology, University College London, London, UK. [31]Department of Clinical Neuroscience, Institute of Neurology, University College London, London, UK. [32]Reta Lila Weston Institute, Institute of Neurology, University College London, London, UK. [33]Department of Neurodegenerative Diseases, Institute of Neurology, University College London, London, UK. [34]Centre for Neuroscience and Trauma, Blizard Institute, Queen Mary University of London, London, UK. [35]North-East London and Essex Regional Motor Neuron Disease Care Centre, London, London, UK. [36]Department of Neurology, Medical School Hannover, Hannover, Germany. [37]Department of Neurology, Otto-von-Guericke University Magdeburg, Magdeburg, Germany. [38]Institute for Clinical Neurobiology, University of Würzburg, Würzburg, Germany. [39]Charité University Hospital, Humboldt-University, Berlin, Germany. [40]Department of Neurology, University of California, San Francisco, California, USA. [41]Center for Neurodegenerative Disease Research, Perelman School of Medicine at the University of Pennsylvania, Philadelphia, Pennsylvania, USA. [42]Department of Neurology, Perelman School of Medicine at the University of Pennsylvania, Pennsylvania, USA. [43]Neurodegeneration Research Laboratory, Bogazici University, Istanbul, Turkey. [44]Institute of Human Genetics, Helmholtz Zentrum München, Neuherberg, Germany. [45]INSERM U930, Université François Rabelais, Tours, France. [46]APHP, Département de Pharmacologie Clinique, Hôpital de la Pitié-Salpêtrière, UPMC Pharmacologie, Paris 6, Paris, France. [47]Department of Neurology, Ulm University, Ulm, Germany. [48]INSERM U 1127, CNRS UMR 7225, Sorbonne Universités, Paris, France. [49]Genethon, CNRS UMR 8587 Evry, France.

[50]Department of Medical Genetics, L'Institut du Cerveau et de la Moelle Épinière, Hoptial Salpêtrière, Paris. [51]Department of Neurogenetics, Institute of Neurology, University College London, London, UK. [52]PopGen Biobank and Institute of Epidemiology, Christian Albrechts-University Kiel, Kiel, Germany. [53]Institute of Clinical Molecular Biology, Kiel University, Kiel, Germany. [54]Central Institute of Mental Health, Mannheim, Germany; Medical Faculty Mannheim. [55]Institute of Human Genetics, University of Bonn, Bonn, Germany. [56]Department of Genomics, Life and Brain Center, Bonn, Germany. [57]University Hospital Basel, University of Basel, Basel, Switzerland. [58]Division of Medical Genetics, Department of Biomedicine, University of Basel, Basel, Switzerland. [59]Institute of Neuroscience and Medicine INM-1, Research Center Juelich, Juelich, Germany. [60]Lille University, INSERM U744, Institut Pasteur de Lille, Lille, France. [61]Bordeaux University, ISPED, Centre INSERM U897-Epidemiologie-Biostatistique & CIC-1401, CHU de Bordeaux, Pole de Sante Publique, Bordeaux, France. [62]Department of Internal Medicine, Genetics Laboratory, Erasmus Medical Center Rotterdam, Rotterdam, The Netherlands. [63]Department of Epidemiology, Erasmus Medical Center Rotterdam, Rotterdam, The Netherlands. [64]MRC Social, Genetic and Developmental Psychiatry Centre, King's College London, London, London, UK. [65]Neuromuscular Diseases Unit/ALS Clinic, Kantonsspital St Gallen, 9007 St Gallen, Switzerland. [66]Laboratory of Experimental Neurobiology, IRCCS 'C. Mondino' National Institute of Neurology Foundation, Pavia, Italy. [67]Neurologic Unit, IRCCS Foundation Ca' Granda Ospedale Maggiore Policlinico, Milan, Italy. [68]Department of Health Sciences, Interdisciplinary Research Center of Autoimmune Diseases, UPO, Università del Piemonte Orientale, Novara, Italy. [69]Department of Neurosciences, University of Padova, Padova, Italy. [70]Department of Neurology, University of Eastern Piedmont, Novara, Italy. [71]Unit of Genetics of Neurodegenerative and Metabolic Diseases, Fondazione IRCCS Istituto Neurologico 'Carlo Besta', Milano, Italy. [72]'Rita Levi Montalcini' Department of Neuroscience, ALS Centre, University of Torino, Turin, Italy. [73]Azienda Ospedaliera Città della Salute e della Scienza, Torino, Italy. [74]Department of Clinical research in Neurology, University of Bari 'A. Moro', at Pia Fondazione 'Card. G. Panico', Tricase, Italy. [75]NEMO Clinical Center, Serena Onlus Foundation, Niguarda Ca' Granda Hostipal, Milan, Italy. [76]Medical Genetics Unit, Department of Laboratory Medicine, Niguarda Ca' Granda Hospital, Milan, Italy. [77]Department of Neurology, Institute of Experimental Neurology (INSPE), Division of Neuroscience, San Raffaele Scientific Institute, Milan, Italy. [78]University Hospital 'Spedali Civili', Brescia, Italy. [79]Department of Neurology, Brighton and Sussex Medical School Trafford Centre for Biomedical Research, University of Sussex, Falmer, East Sussex, UK. [80]Laboratory of Neurological Diseases, Department of Neuroscience, IRCCS Istituto di Ricerche Farmacologiche Mario Negri, Milano, Italy. [81]Department of Basic Medical Sciences, Neuroscience and Sense Organs, University of Bari 'Aldo Moro', Bari, Italy. [82]Unit of Neurodegenerative Diseases, Department of Clinical Research in Neurology, University of Bari 'Aldo Moro', at Pia Fondazione Cardinale G. Panico, Tricase, Lecce, Italy. [83]Department of Neurology, University Hospital Leuven, Leuven Belgium. [84]KU Leuven-University of Leuven, Department of Neurosciences, VIB-Vesalius Research Center, Leuven, Belgium. [85]Department of Neurology, University of Massachusetts Medical School, Worcester, Massachusetts, USA. [86]Department of Pharmacology and Clinical Neurosience, Umeå University, Umeå, Sweden. [87]Centre SLA, CHRU de Tours, Tours, France. [88]Federation des Centres SLA Tours and Limoges, LITORALS, Tours, France. [89]Department of Translational Neuroscience, Brain Center Rudolf Magnus, University Medical Center Utrecht, Utrecht, The Netherlands. [90]Department of Genetics, University of Groningen, University Medical Centre Groningen, Groningen, The Netherlands. [91]Department of Medical and Molecular Genetics, King's College London, London, UK. [92]IoPPN Genomics & Biomarker Core, Translational Genetics Group, MRC Social, Genetic and Developmental Psychiatry Centre, King's College London, London, UK. [93]NIHR Biomedical Research Centre for Mental Health, Maudsley Hospital and Institute of Psychiatry, Psychology & Neuroscience, King's College London, London, UK.

## Schizophrenia Working Group of the Psychiatric Genomics Consortium

Stephan Ripke[94,95], Benjamin M. Neale[94,95,96,97], Aiden Corvin[98], James T.R. Walters[99], Kai-How Farh[94], Peter A. Holmans[99,100], Phil Lee[94,95,97], Brendan Bulik-Sullivan[94,95], David A. Collier[101,102], Hailiang Huang[94,96], Tune H. Pers[96,103,104], Ingrid Agartz[105,106,107], Esben Agerbo[108,109,110], Margot Albus[111], Madeline Alexander[112], Farooq Amin[113,114], Silviu A. Bacanu[115], Martin Begemann[116], Richard A. Belliveau Jr[95], Judit Bene[117,118], Sarah E. Bergen[95,119], Elizabeth Bevilacqua[95], Tim B. Bigdeli[115], Donald W. Black[120], Richard Bruggeman[121], Nancy G. Buccola[122], Randy L. Buckner[123,124,125], William Byerley[126], Wiepke Cahn[4], Guiqing Cai[127,128], Dominique Campion[129], Rita M. Cantor[5], Vaughan J. Carr[130,131], Noa Carrera[99], Stanley V. Catts[130,132], Kimberley D. Chambert[95], Raymond C.K. Chan[133], Ronald Y.L. Chan[134], Eric Y.H. Chen[134,135], Wei Cheng[136], Eric F.C. Cheung[137], Siow Ann Chong[138], C. Robert Cloninger[139], David Cohen[140], Nadine Cohen[141], Paul Cormican[98], Nick Craddock[99,100], James J. Crowley[142], David Curtis[143,144], Michael Davidson[145], Kenneth L. Davis[128], Franziska Degenhardt[55,56], Jurgen Del Favero[146], Ditte Demontis[110,147,148], Dimitris Dikeos[149], Timothy Dinan[150], Srdjan Djurovic[107,151], Gary Donohoe[98,152], Elodie Drapeau[128], Jubao Duan[153,154], Frank Dudbridge[155], Naser Durmishi[156], Peter Eichhammer[157], Johan Eriksson[158,159,160], Valentina Escott-Price[99], Laurent Essioux[161], Ayman H. Fanous[162,163,164,165], Martilias S. Farrell[142], Josef Frank[166], Lude Franke[90], Robert Freedman[167], Nelson B. Freimer[6], Marion Friedl[168], Joseph I. Friedman[128], Menachem Fromer[94,95,97,169], Giulio Genovese[95], Lyudmila Georgieva[99], Ina Giegling[168,170], Paola Giusti-Rodríguez[142], Stephanie Godard[171], Jacqueline I. Goldstein[94,96], Vera Golimbet[172], Srihari Gopal[141], Jacob Gratten[173], Lieuwe de Haan[174], Christian Hammer[116], Marian L. Hamshere[99], Mark Hansen[175], Thomas Hansen[110,176], Vahram Haroutunian[128,177,178], Annette M. Hartmann[168], Frans A. Henskens[130,179,180], Stefan Herms[55,56,58], Joel N. Hirschhorn[96,104,181], Per Hoffmann[55,56,58], Andrea Hofman[55,56], Mads V. Hollegaard[182], David M. Hougaard[182], Masashi Ikeda[183], Inge Joa[184], Antonio Julià[185], Luba Kalaydjieva[186,187], Sena Karachanak-Yankova[188], Juha Karjalainen[90], David Kavanagh[99],

Matthew C. Keller[189], James L. Kennedy[190,191,192], Andrey Khrunin[193], Yunjung Kim[142], Janis Klovins[194], James A. Knowles[195], Bettina Konte[168], Vaidutis Kucinskas[196], Zita Ausrele Kucinskiene[196], Hana Kuzelova-Ptackova[197,198], Anna K. Kähler[119], Claudine Laurent[112,199], Jimmy Lee[138,200], S. Hong Lee[173], Sophie E. Legge[99], Bernard Lerer[201], Miaoxin Li[134,202], Tao Li[203], Kung-Yee Liang[204], Jeffrey Lieberman[205], Svetlana Limborska[193], Carmel M. Loughland[130,206], Jan Lubinski[207], Jouko Lönnqvist[208], Milan Macek[197,198], Patrik K.E. Magnusson[119], Brion S. Maher[209], Wolfgang Maier[210], Jacques Mallet[211], Sara Marsal[185], Manuel Mattheisen[110,147,148,212], Morten Mattingsdal[107,213], Robert W. McCarley[214,215], Colm McDonald[216], Andrew M. McIntosh[217,218], Sandra Meier[166], Carin J. Meijer[174], Bela Melegh[117,118], Ingrid Melle[107,219], Raquelle I. Mesholam-Gately[214,220], Andres Metspalu[221], Patricia T. Michie[130,222], Lili Milani[221], Vihra Milanova[223], Younes Mokrab[101], Derek W. Morris[98,152], Ole Mors[110,147,224], Kieran C. Murphy[225], Robin M. Murray[226], Inez Myin-Germeys[227], Bertram Müller-Myhsok[228,229,230], Mari Nelis[221], Igor Nenadic[231], Deborah A. Nertney[232], Gerald Nestadt[233], Kristin K. Nicodemus[234], Liene Nikitina-Zake[194], Laura Nisenbaum[235], Annelie Nordin[236], Eadbhard O'Callaghan[237], Colm O'Dushlaine[95], F. Anthony O'Neill[238], Sang-Yun Oh[239], Ann Olincy[167], Line Olsen[110,176], Jim Van Os[227,240], Christos Pantelis[130,241], George N. Papadimitriou[149], Sergi Papiol[116], Elena Parkhomenko[128], Michele T. Pato[195], Tiina Paunio[242,243], Milica Pejovic-Milovancevic[244], Diana O. Perkins[245], Olli Pietiläinen[243,246], Jonathan Pimm[144], Andrew J. Pocklington[99], Alkes Price[247], Ann E. Pulver[233], Shaun M. Purcell[169], Digby Quested[248], Henrik B. Rasmussen[110,176], Abraham Reichenberg[128], Mark A. Reimers[249], Alexander L. Richards[99,100], Joshua L. Roffman[123,125], Panos Roussos[169,250], Douglas M. Ruderfer[169], Veikko Salomaa[160], Alan R. Sanders[153,154], Ulrich Schall[130,206], Christian R. Schubert[251], Thomas G. Schulze[166,252], Sibylle G. Schwab[253], Edward M. Scolnick[95], Rodney J. Scott[130,254,255], Larry J. Seidman[214,220], Jianxin Shi[256], Engilbert Sigurdsson[257], Teimuraz Silagadze[258], Jeremy M. Silverman[128,259], Kang Sim[138], Petr Slominsky[193], Jordan W. Smoller[95,97], Hon-Cheong So[134], Chris C. A Spencer[260], Eli A. Stahl[96,169], Hreinn Stefansson[261], Stacy Steinberg[261], Elisabeth Stogmann[262], Richard E. Straub[263], Eric Strengman[264,265], Jana Strohmaier[166], T. Scott Stroup[205], Mythily Subramaniam[138], Jaana Suvisaari[208], Dragan M. Svrakic[139], Jin P. Szatkiewicz[142], Erik Söderman[105], Srinivas Thirumalai[266], Draga Toncheva[188], Sarah Tosato[267], Juha Veijola[268,269], John Waddington[270], Dermot Walsh[271], Dai Wang[141], Qiang Wang[203], Bradley T. Webb[115], Mark Weiser[145], Dieter B. Wildenauer[272], Nigel M. Williams[273], Stephanie Williams[142], Stephanie H. Witt[166], Aaron R. Wolen[249], Emily H.M. Wong[134], Brandon K. Wormley[115], Hualin Simon Xi[274], Clement C. Zai[190,191], Xuebin Zheng[275], Fritz Zimprich[262], Naomi R. Wray[173], Kari Stefansson[261], Peter M. Visscher[173], Rolf Adolfsson[236], Ole A. Andreassen[107,219], Douglas H.R. Blackwood[218], Elvira Bramon[276], Joseph D. Buxbaum[127,128,177,277], Anders D. Børglum[110,147,148,224], Ariel Darvasi[278], Enrico Domenici[279], Hannelore Ehrenreich[116], Tõnu Esko[96,104,181,221], Pablo V. Gejman[153,154], Michael Gill[98], Hugh Gurling[144], Christina M. Hultman[119], Nakao Iwata[183], Assen V. Jablensky[130,280,281,282], Erik G. Jönsson[105], Kenneth S. Kendler[283], George Kirov[99], Jo Knight[190,191,192], Todd Lencz[284,285,286], Douglas F. Levinson[112], Qingqin S. Li[141], Jianjun Liu[275,287], Anil K. Malhotra[284,285,286], Steven A. McCarroll[95,181], Andrew McQuillin[144], Jennifer L. Moran[95], Preben B. Mortensen[108,109,110], Bryan J. Mowry[173,288], Michael J. Owen[99,100], Aarno Palotie[97,246,289], Carlos N. Pato[195], Tracey L. Petryshen[214,289,290], Danielle Posthuma[291,292,293], Brien P. Riley[283], Dan Rujescu[168,170], Pak C. Sham[134,135,202], Pamela Sklar[169,177,250], David St Clair[294], Daniel R. Weinberger[263,295], Jens R. Wendland[251], Thomas Werge[110,176,296], Mark J. Daly[94], Patrick F. Sullivan[119,142,245] & Michael C. O'Donovan[99,100]

[94]Analytic and Translational Genetics Unit, Massachusetts General Hospital, Boston, Massachusetts, USA. [95]Stanley Center for Psychiatric Research, Broad Institute of MI.T. and Harvard, Cambridge, Massachusetts, USA. [96]Medical and Population Genetics Program, Broad Institute of MI.T. and Harvard,

Cambridge, Massachusetts, USA. [97]Psychiatric and Neurodevelopmental Genetics Unit, Massachusetts General Hospital, Boston, Massachusetts, USA. [98]Neuropsychiatric Genetics Research Group, Department of Psychiatry, Trinity College, Dublin, Ireland. [99]MRC Centre for Neuropsychiatric Genetics and Genomics, Institute of Psychological Medicine and Clinical Neurosciences, School of Medicine, Cardiff University, Cardiff, UK. [100]National Centre for Mental Health, Cardiff University, Cardiff, Wales. [101]Eli Lilly and Company Limited, Erl Wood Manor, Sunninghill Road, Windlesham, Surrey, UK. [102]Social, Genetic and Developmental Psychiatry Centre, Institute of Psychiatry, King's College London, London, UK. [103]Center for Biological Sequence Analysis, Department of Systems Biology, Technical University of Denmark, Lyngby, Denmark. [104]Division of Endocrinology and Center for Basic and Translational Obesity Research, Boston Children's Hospital, Boston, Massachusetts, USA. [105]Department of Clinical Neuroscience, Karolinska Institutet, Stockholm, Sweden. [106]Department of Psychiatry, Diakonhjemmet Hospital, Oslo, Norway. [107]NORMENT, K.G. Jebsen Centre for Psychosis Research, Institute of Clinical Medicine, University of Oslo, Oslo, Norway. [108]Centre for Integrative Register-based Research, CIRRAU, Aarhus University, Aarhus, Denmark. [109]National Centre for Register-based Research, Aarhus University, Aarhus, Denmark. [110]The Lundbeck Foundation Initiative for Integrative Psychiatric Research, iPSYCH, Denmark. [111]State Mental Hospital, Haar, Germany. [112]Department of Psychiatry and Behavioral Sciences, Stanford University, Stanford, California, USA. [113]Department of Psychiatry and Behavioral Sciences, Atlanta Veterans Affairs Medical Center, Atlanta, Georgia, USA. [114]Department of Psychiatry and Behavioral Sciences, Emory University, Atlanta, Georgia, USA. [115]Virginia Institute for Psychiatric and Behavioral Genetics, Department of Psychiatry, Virginia Commonwealth University, Richmond, Virginia, USA. [116]Clinical Neuroscience, Max Planck Institute of Experimental Medicine, Göttingen, Germany. [117]Department of Medical Genetics, University of Pécs, Pécs, Hungary. [118]Szentagothai Research Center, University of Pécs, Pécs, Hungary. [119]Department of Medical Epidemiology and Biostatistics, Karolinska Institutet, Stockholm, Sweden. [120]Department of Psychiatry, University of Iowa Carver College of Medicine, Iowa City, Iowa, USA. [121]University Medical Center Groningen, Department of Psychiatry, University of Groningen, The Netherlands. [122]School of Nursing, Louisiana State University Health Sciences Center, New Orleans, Louisiana, USA. [123]Athinoula A. Martinos Center, Massachusetts General Hospital, Boston, Massachusetts, USA. [124]Center for Brain Science, Harvard University, Cambridge Massachusetts, USA. [125]Department of Psychiatry, Massachusetts General Hospital, Boston, Massachusetts, USA. [126]Department of Psychiatry, University of California at San Francisco, San Francisco, California, USA. [127]Department of Human Genetics, Icahn School of Medicine at Mount Sinai, New York, New York, USA. [128]Department of Psychiatry, Icahn School of Medicine at Mount Sinai, New York, New York, USA. [129]Centre Hospitalier du Rouvray and INSER.M. U1079 Faculty of Medicine, Rouen, France. [130]Schizophrenia Research Institute, Sydney, Australia. [131]School of Psychiatry, University of New South Wales, New South Wales, Sydney, Australia. [132]Royal Brisbane and Women's Hospital, University of Queensland, Queensland, Brisbane, Australia. [133]Institute of Psychology, Chinese Academy of Science, Beijing, China. [134]Department of Psychiatry, Li Ka Shing Faculty of Medicine, The University of Hong Kong, Hong Kong, China. [135]State Ket Laboratory for Brain and Cognitive Sciences, Li Ka Shing Faculty of Medicine, The University of Hong Kong, Hong Kong, China. [136]Department of Computer Science, University of North Carolina, Chapel Hill, North Carolina, USA. [137]Castle Peak Hospital, Hong Kong, China. [138]Institute of Mental Health, Singapore. [139]Department of Psychiatry, Washington University, St Louis, Missouri, USA. [140]Department of Child and Adolescent Psychiatry, Pierre and Marie Curie Faculty of Medicine and Brain and Spinal Cord Institute (ICM), Paris, France. [141]Neuroscience Therapeutic Area, Janssen Research and Development, LLC, Raritan, New Jersey, USA. [142]Department of Genetics, University of North Carolina, Chapel Hill, North Carolina, USA. [143]Department of Psychological Medicine, Queen Mary University of London, London, UK. [144]Molecular Psychiatry Laboratory, Division of Psychiatry, University College London, London, UK. [145]Sheba Medical Center, Tel Hashomer, Israel. [146]Applied Molecular Genomics Unit, VI.B. Department of Molecular Genetics, University of Antwerp, Antwerp, Belgium. [147]Centre for Integrative Sequencing, iSEQ, Aarhus University, Aarhus, Denmark. [148]Department of Biomedicine, Aarhus University, Aarhus, Denmark. [149]First Department of Psychiatry, University of Athens Medical School, Athens, Greece. [150]Department of Psychiatry, University College Cork, Ireland. [151]Department of Medical Genetics, Oslo University Hospital, Oslo, Norway. [152]Cognitive Genetics and Therapy Group, School of Psychology and Discipline of Biochemistry, National University of Ireland Galway, Ireland. [153]Department of Psychiatry and Behavioral Neuroscience, University of Chicago, Chicago, Illinois, USA. [154]Department of Psychiatry and Behavioral Sciences, NorthShore University HealthSystem, Evanston, Illinois, USA. [155]Department of Non-Communicable Disease Epidemiology, London School of Hygiene and Tropical Medicine, London, London, UK. [156]Department of Child and Adolescent Psychiatry, University Clinic of Psychiatry, Skopje, Republic of Macedonia. [157]Department of Psychiatry, University of Regensburg, Regensburg, Germany. [158]Department of General Practice, Helsinki University Central Hospital, Helsinki, Finland. [159]Folkhälsan Research Center, Helsinki, Finland. [160]National Institute for Health and Welfare, Helsinki, Finland. [161]Translational Technologies and Bioinformatics, Pharma Research and Early Development, F. Hoffman-La Roche, Basel, Switzerland. [162]Department of Psychiatry, Georgetown University School of Medicine, Washington, District Of Columbia, USA. [163]Department of Psychiatry, Keck School of Medicine of the University of Southern California, Los Angeles, California, USA. [164]Department of Psychiatry, Virginia Commonwealth University School of Medicine, Richmond, Virginia, USA. [165]Mental Health Service Line, Washington V.A. Medical Center, Washington, District Of Columbia, USA. [166]Department of Genetic Epidemiology in Psychiatry, Central Institute of Mental Health, Medical Faculty Mannheim, University of Heidelberg, Heidelberg, Germany. [167]Department of Psychiatry, University of Colorado Denver, Aurora, Colorado, USA. [168]Department of Psychiatry, University of Halle, Halle, Germany. [169]Division of Psychiatric Genomics, Department of Psychiatry, Icahn School of Medicine at Mount Sinai, New York, New York, USA. [170]Department of Psychiatry, University of Munich, Munich, Germany. [171]Departments of Psychiatry and Human and Molecular Genetics, INSERM, Institut de Myologie, Hôpital de la Pitiè-Salpêtrière, Paris, France. [172]Mental Health Research Centre, Russian Academy of Medical Sciences, Moscow, Russia. [173]Queensland Brain Institute, The University of Queensland, Brisbane, Queensland, Australia. [174]Academic Medical Centre University of Amsterdam, Department of Psychiatry, Amsterdam, The Netherlands. [175]Illumina, Inc., La Jolla, California, USA. [176]Institute of Biological Psychiatry, MH.C. Sct. Hans, Mental Health Services, Copenhagen, Denmark. [177]Friedman Brain Institute, Icahn School of Medicine at Mount Sinai, New York, New York, USA. [178]JJ Peters V.A. Medical Center, Bronx, New York, USA. [179]Priority Research Centre for Health Behaviour, University of Newcastle, Newcastle, Australia. [180]School of Electrical Engineering and Computer Science, University of Newcastle, Newcastle, Australia. [181]Department of Genetics, Harvard Medical School, Boston, Massachusetts, USA. [182]Section of Neonatal Screening and Hormones, Department of Clinical Biochemistry, Immunology and Genetics, Statens Serum Institut, Copenhagen, Denmark. [183]Department of Psychiatry, Fujita Health University School of Medicine, Toyoake, Aichi, Japan. [184]Regional Centre for Clinical Research in Psychosis, Department of Psychiatry, Stavanger University Hospital, Stavanger, Norway. [185]Rheumatology Research Group, Vall d'Hebron Research Institute, Barcelona, Spain. [186]Centre for Medical Research, The University of Western Australia, Perth, Western Australia, Australia. [187]Perkins Institute for Medical Research, The University of Western Australia, Perth, Western Australia, Australia. [188]Department of Medical Genetics, Medical University, Sofia, Bulgaria. [189]Department of Psychology, University of Colorado Boulder, Boulder, Colorado, USA. [190]Campbell Family Mental Health Research Institute, Centre for Addiction and Mental Health, Toronto, Ontario, Canada. [191]Department of Psychiatry, University of Toronto, Toronto, Ontario, Canada. [192]Institute of Medical Science, University of Toronto, Toronto, Ontario, Canada. [193]Institute of Molecular Genetics, Russian Academy of Sciences, Moscow, Russia. [194]Latvian Biomedical Research and Study Centre, Riga, Latvia. [195]Department of Psychiatry and Zilkha Neurogenetics Institute, Keck School of Medicine at University of Southern California, Los Angeles, California, USA. [196]Faculty of Medicine, Vilnius University, Vilnius, Lithuania. [197]Second Faculty of Medicine and University Hospital Motol, Prague, Czech Republic. [198]Department of Biology and Medical Genetics, Charles University Prague, Prague, Czech Republic. [199]Pierre and Marie Curie Faculty of Medicine, Paris, France. [200]Duke-NUS Graduate Medical School, Singapore. [201]Department of Psychiatry, Hadassah-Hebrew University Medical Center, Jerusalem, Israel. [202]Centre for Genomic Sciences, Li Ka Shing Faculty of Medicine, The University of Hong Kong, Hong Kong, China. [203]Mental Health Centre and Psychiatric Laboratory, West China Hospital, Sichuan University, Chendu, Sichuan, China. [204]Department of Biostatistics, Johns Hopkins University Bloomberg School of Public Health, Baltimore, Maryland, USA. [205]Department of Psychiatry, Columbia University, New York,

New York, USA. [206]Priority Centre for Translational Neuroscience and Mental Health, University of Newcastle, Newcastle, Australia. [207]Department of Genetics and Pathology, International Hereditary Cancer Center, Pomeranian Medical University in Szczecin, Szczecin, Poland. [208]Department of Mental Health and Substance Abuse Services, National Institute for Health and Welfare, Helsinki, Finland. [209]Department of Mental Health, Bloomberg School of Public Health, Johns Hopkins University, Baltimore, Maryland, USA. [210]Department of Psychiatry, University of Bonn, Bonn, Germany. [211]Centre National de la Recherche Scientifique, Laboratoire de Génétique Moléculaire de la Neurotransmission et des Processus Neurodégénératifs, Hôpital de la Pitié Salpêtrière, Paris, France. [212]Department of Genomics Mathematics, University of Bonn, Bonn, Germany. [213]Research Unit, Sørlandet Hospital, Kristiansand, Norway. [214]Department of Psychiatry, Harvard Medical School, Boston, Massachusetts, USA. [215]Virginia Boston Health Care System, Brockton, Massachusetts, USA. [216]Department of Psychiatry, National University of Ireland Galway, Ireland. [217]Centre for Cognitive Ageing and Cognitive Epidemiology, University of Edinburgh, Edinburgh, UK. [218]Division of Psychiatry, University of Edinburgh, Edinburgh, UK. [219]Division of Mental Health and Addiction, Oslo University Hospital, Oslo, Norway. [220]Massachusetts Mental Health Center Public Psychiatry Division of the Beth Israel Deaconess Medical Center, Boston, Massachusetts, USA. [221]Estonian Genome Center, University of Tartu, Tartu, Estonia. [222]School of Psychology, University of Newcastle, Newcastle, Australia. [223]First Psychiatric Clinic, Medical University, Sofia, Bulgaria. [224]Department P, Aarhus University Hospital, Risskov, Denmark. [225]Department of Psychiatry, Royal College of Surgeons in Ireland, Ireland. [226]King's College London, London, UK. [227]Maastricht University Medical Centre, South Limburg Mental Health Research and Teaching Network, EURON, Maastricht, The Netherlands. [228]Institute of Translational Medicine, University Liverpool, UK. [229]Max Planck Institute of Psychiatry, Munich, Germany. [230]Munich Cluster for Systems Neurology (SyNergy), Munich, Germany. [231]Department of Psychiatry and Psychotherapy, Jena University Hospital, Jena, Germany. [232]Department of Psychiatry, Queensland Brain Institute and Queensland Centre for Mental Health Research, University of Queensland, Brisbane, Queensland, Australia. [233]Department of Psychiatry and Behavioral Sciences, Johns Hopkins University School of Medicine, Baltimore, Maryland, USA. [234]Department of Psychiatry, Trinity College Dublin, Dublin, Ireland. [235]Eli Lilly and Company, Lilly Corporate Center, Indianapolis, Indiana, USA. [236]Department of Clinical Sciences, Psychiatry, Umeå University, Umeå, Sweden. [237]DETECT Early Intervention Service for Psychosis, Blackrock, Dublin, Ireland. [238]Centre for Public Health, Institute of Clinical Sciences, Queens University Belfast, Belfast, UK. [239]Lawrence Berkeley National Laboratory, University of California at Berkeley, Berkeley, California, USA. [240]Institute of Psychiatry at King's College London, London, UK. [241]Melbourne Neuropsychiatry Centre, University of Melbourne & Melbourne Health, Melbourne, Australia. [242]Department of Psychiatry, University of Helsinki, Finland. [243]Public Health Genomics Unit, National Institute for Health and Welfare, Helsinki, Helsinki, Finland. [244]Medical Faculty, University of Belgrade, Belgrade, Serbia. [245]Department of Psychiatry, University of North Carolina, Chapel Hill, North Carolina, USA. [246]Institute for Molecular Medicine Finland, FIMM, Helsinki, Finland. [247]Department of Epidemiology, Harvard University, Boston, Massachusetts, USA. [248]Department of Psychiatry, University of Oxford, Oxford, UK. [249]Virginia Institute for Psychiatric and Behavioral Genetics, Virginia Commonwealth University, Richmond, Virginia, USA. [250]Institute for Multiscale Biology, Icahn School of Medicine at Mount Sinai, New York, New York, USA. [251]PharmaTherapeutics Clinical Research, Pfizer Worldwide Research and Development, Cambridge, Massachusetts, USA. [252]Department of Psychiatry and Psychotherapy, University of Gottingen, Göttingen, Germany. [253]Psychiatry and Psychotherapy Clinic, University of Erlangen, Erlangen, Germany. [254]Hunter New England Health Service, Newcastle, Australia. [255]School of Biomedical Sciences, University of Newcastle, Newcastle, Australia. [256]Division of Cancer Epidemiology and Genetics, National Cancer Institute, Bethesda, Maryland, USA. [257]University of Iceland, Landspitali, National University Hospital, Reykjavik, Iceland. [258]Department of Psychiatry and Drug Addiction, Tbilisi State Medical University (TSMU), Tbilisi, Georgia. [259]Research and Development, Bronx Veterans Affairs Medical Center, New York, New York, USA. [260]Wellcome Trust Centre for Human Genetics, Oxford, UK. [261]deCODE Genetics, Reykjavik, Iceland. [262]Department of Clinical Neurology, Medical University of Vienna, Vienna, Austria. [263]Lieber Institute for Brain Development, Baltimore, Maryland, USA. [264]Department of Medical Genetics, University Medical Centre, Utrecht, The Netherlands. [265]Rudolf Magnus Institute of Neuroscience, University Medical Centre Utrecht, Utrecht, The Netherlands. [266]Berkshire Healthcare NH.S. Foundation Trust, Bracknell, UK. [267]Section of Psychiatry, University of Verona, Verona, Italy. [268]Department of Psychiatry, University of Oulu, Oulu, Finland. [269]University Hospital of Oulu, Oulu, Finland. [270]Molecular and Cellular Therapeutics, Royal College of Surgeons in Ireland, Dublin, Ireland. [271]Health Research Board, Dublin, Ireland. [272]Department of Psychiatry and Clinical Neurosciences, School of Psychiatry and Clinical Neurosciences, Queen Elizabeth I.I. Medical Centre, Perth, Western Australia, Australia. [273]Department of Psychological Medicine and Neurology, MR.C. Centre for Neuropsychiatric Genetics and Genomics, School of Medicine, Cardiff University, Cardiff, Wales, UK. [274]Computational Sciences CoE, Pfizer Worldwide Research and Development, Cambridge, Massachusetts, USA. [275]Human Genetics, Genome Institute of Singapore, Singapore. [276]University College London, London, UK. [277]Department of Neuroscience, Icahn School of Medicine at Mount Sinai, New York, New York, USA. [278]Department of Genetics, The Hebrew University of Jerusalem, Jerusalem, Israel. [279]Neuroscience Discovery and Translational Area, Pharma Research and Early Development, F. Hoffman-La Roche, Basel, Switzerland. [280]School of Psychiatry and Clinical Neurosciences, The University of Western Australia, Perth, Australia. [281]The Perkins Institute of Medical Research, Perth, Australia. [282]UWA Centre for Clinical Research in Neuropsychiatry. [283]Virginia Institute for Psychiatric and Behavioral Genetics, Departments of Psychiatry and Human and Molecular Genetics, Virginia Commonwealth University, Richmond, Virginia, USA. [284]The Feinstein Institute for Medical Research, Manhasset, New York, USA. [285]The Hofstra NS-LIJ School of Medicine, Hempstead, New York, USA. [286]The Zucker Hillside Hospital, Glen Oaks, New York, USA. [287]Saw Swee Hock School of Public Health, National University of Singapore, Singapore. [288]Queensland Centre for Mental Health Research, University of Queensland, Brisbane, Queensland, Australia. [289]The Broad Institute of MI.T. and Harvard, Cambridge, Massachusetts, USA. [290]Center for Human Genetic Research and Department of Psychiatry, Massachusetts General Hospital, Boston, Massachusetts, USA. [291]Department of Child and Adolescent Psychiatry, Erasmus University Medical Centre, Rotterdam, The Netherlands. [292]Department of Complex Trait Genetics, Neuroscience Campus Amsterdam, V.U. University Medical Center Amsterdam, Amsterdam, The Netherlands. [293]Department of Functional Genomics, Center for Neurogenomics and Cognitive Research, Neuroscience Campus Amsterdam, VU University, Amsterdam, The Netherlands. [294]University of Aberdeen, Institute of Medical Sciences, Aberdeen, Scotland, UK. [295]Departments of Psychiatry, Neurology, Neuroscience and Institute of Genetic Medicine, Johns Hopkins School of Medicine, Baltimore, Maryland, USA. [296]Department of Clinical Medicine, University of Copenhagen, Copenhagen, Denmark.

