## [Peer Review File · Nature Communications]

Reviewers' Comments:

Reviewer #1 (Remarks to the Author)

McLaughlin and colleagues have performed a thorough study examining the genetic overlap between amyotrophic lateral sclerosis (ALS) and schizophrenia. They examined genome-wide association study (GWAS) datasets from 36,052 individuals in an ALS cohort and 79,845 individuals in a schizophrenia cohort. Using linkage disequilibrium (LD) score regression they demonstrated an overlap of 14.3% between ALS and schizophrenia, and with polygenic risk scores (PRS) they explained up to 0.12% of the variance in ALS. Their results seem to indicate that this overlap is specific to schizophrenia, since a similar overlap was not observed when comparing their cohort of ALS patients to cohorts of patients with Alzheimer's disease (AD), attention deficit-hyperactivity disorder (ADHD), autism spectrum disorder (ASD), bipolar disorder (BPD), major depressive disorder (MDD), or multiple sclerosis (MS). Additionally, they identified new potential loci that were shared between patients with ALS and schizophrenia (CNTN6, TNIP1, PPP2R2D, NCKAP5L, and ZNF295-AS1). The findings described by McLaughlin and colleagues are interesting, and this is a clear, concise, and well-written manuscript.

Comments:

- Abstract:

o Even though findings can be significant, they do not have to be clinically meaningful. A genetic correlation between ALS and schizophrenia of 14.3% seems reasonable, but since the PRS only explains up to 0.12% of the variance in ALS, one could wonder whether this relatively small percentage is clinically meaningful. Maybe the authors could comment on the relevance of their findings?

o Because only 0.12% of the variance in ALS appears to be explained by schizophrenia PRS, the statement that "shared neurobiological mechanisms between these two disorders will be relevant to future preclinical and clinical studies and in novel drug discovery" seems too strong. Please adjust accordingly.

- Introduction:

o Maybe the authors could elaborate on how they define the additive polygenic risk conferred by common genetic variants? Their definition is probably similar to the definition used in large-scale studies focusing on other diseases, but quite different from the definition generally used in the ALS field. To assist readers it would help to include a definition, especially since the authors are using summary data (at least for a subset of samples and analyses), and consequently, it is important for readers to realize that they are not referring to the presence of multiple variants within a single individual.

o In the last sentence of their introduction, the authors mention that they "demonstrate that this polygenic overlap with ALS is specific to schizophrenia". This statement seems too strong, especially since the number of subjects examined with other diseases might have been too small (limited power). Please see the Results section for more detailed comments.

o It might be useful to include a brief description of LD score regression upon its first introduction in the text. For instance, that LD score regression is used because it is able to distinguish confounding from polygenicity, which makes sense since "inflation from cryptic relatedness within or between cohort or population stratification purely from genetic drift will not correlate with LD" (Bulik-Sullivan et al., Nature Genetics, 2015). Variants in LD with a causal variant will show an elevation in test statistics in association analysis proportional to their LD with the causal variant.

- Results:

o The authors show that the schizophrenia-based PRS accounts for 0.12% of the phenotypic variance in the ALS patients. Are the authors able to test the opposite, determine the ALS-based

PRS in schizophrenia patients?

o The authors included information about Breaking Up Heterogeneous Mixture Based On Cross-locus correlations (BUHMBOX) in the supplementary information, but it might be useful to include a concise description in the main text, including a definition for heterogeneity (e.g., one disease caused by different mutations) and for pleiotropy (e.g., one mutation causing different diseases).

o They stress that if the genetic overlap between ALS and schizophrenia is driven by misdiagnosis of ALS as schizophrenia, then the percentage of misdiagnosis has to be around 5%. ALS and schizophrenia are clinically very different and 5% does seem unlikely; however, since a subset of ALS patients demonstrates signs of frontal temporal dementia (FTD; for example, behavioral variant FTD [bvFTD]), one cannot exclude the possibility that some of those cases who developed symptoms of FTD, possibly before they exhibited symptoms of ALS, could be misdiagnosed with schizophrenia. Although it remains unlikely that misdiagnosis explains the entire overlap (especially because of the difference in age at onset), it could, potentially, explain a proportion of the overlap.

o It is unclear why the authors did not put more emphasis on the potential novel loci. It would be interesting, for instance, to investigate whether there is an increased burden of rare variants in identified genes (e.g., CNTN6, TNIP1, PPP2R2D, NCKAP5L, and ZNF295-AS1). Moreover, please provide a brief explanation why a conditional false discovery rate (cFDR) of less than 0.01 was used.

o The authors write that "to investigate the polygenic overlap between ALS and schizophrenia, we used individual-level and summary data from GWAS for ALS ... and schizophrenia ..." Maybe the authors could make it easier for readers to understand for which analyses they used individual-level data and for which analyses summary data?

o It is confusing that the authors refer to Supplementary Information 2, before they refer to Supplementary Information 1. Please put the Supplementary Information in order.

o Based on their results, it does not seem fair to conclude that the genetic correlation between ALS and schizophrenia is specific to schizophrenia. Figure 1 demonstrates substantial error bars for most comparisons, which might be a reflection of the relatively small sample size of some studies (e.g., 5,422 individuals in the ADHD cohort). The effect of sample size is emphasized when examining the height variable: there is no reason to assume that height demonstrates a significant correlation with ALS, but because of the relatively large sample size (253,288) even a small difference almost reaches significance. In their Discussion, the authors do mention that the absence of an overlap "could relate to statistical power conferred by secondary phenotype cohort sizes", which is correct. They should, therefore, be more careful with the interpretation of their findings.

- Discussion:

o The Discussion contains statements without references. Please add additional references.

o As mentioned previously, it is true that it seems unlikely that misdiagnosis explains the majority of the overlap, but the authors cannot exclude the possibility that misdiagnosis might account for a (small) proportion of the overlap.

- Methods (including Supplementary Information):

o McLaughlin and colleagues constrained the intercept of their mixed linear model (MLM), because they wanted to be conservative. It has been reported, however, that "constrained intercept LD score regression has lower standard error - often by as much as 30% - than LD score regression with unconstrained intercept, but will yield biased and misleading estimates if the intercept is misspecified, e.g., if we ... do not completely control for population stratification" (Bulik-Sullivan et al., Nature Genetics, 2015). Maybe the authors could comment on this and explain how their results would change if they would not use a constrained model?

o Since the authors also identified five loci already known to be involved in ALS, including C9ORF72 and TANK binding kinase 1 (TBK1), it seems that they have not excluded patients with known mutations, nor have they removed (other) loci known to be involved in ALS. It would be interesting to perform a sensitivity analysis to determine what happens after exclusion of known loci. How would this sensitivity analysis influence the SNP-based heritability, the genetic

correlation, and PRS? This is particularly relevant because of reports that have described patients with a repeat expansion in C9ORF72 with atypical symptoms, such as schizophrenia and psychotic illnesses. It would also ensure that the genetic correlation is not driven by a single locus (or a few loci) or biased by the fit of the model (Hagenaars et al., *Molecular Psychiatry*, 2016). Have the authors considered removing all genome-wide significant loci?

o It looks as if the supplementary data is formatted for another journal, please use "." instead of ".,".

o It is unclear whether subjects excluded from the ALS GWAS from the Finnish and German strata were present in the schizophrenia group or in the control group of the schizophrenia cohort. Why were those subjects not excluded from the schizophrenia dataset instead of from the ALS dataset, since the ALS dataset is smaller? Due to the lack of individual-level data for a subset of samples?

o The Supplementary Information is very helpful and complete, and the legends are brief, but informative. Sometimes, however, the order of the rows in tables varies, which can be confusing. For instance, Supplementary Tables 1 and 2 start with MLM constrained, but Supplementary Table 7 starts with MLM free.

- Typos:

o Occasionally, references are not inserted properly, e.g., "cohort-level logistic regression12." instead of "cohort-level logistic regression.12"

o The authors already use "SNP" in their Introduction, but the abbreviation is introduced in the Results section. Please introduce all abbreviations upon their first occurrence in the text.

Reviewer #2 (Remarks to the Author)

1. This study did not show the details of estimated heritability calculated by using the LDSC. A heritability Z score of > 0.4 and a mean χ^2 statistic of > 1.02 were usually used to ensure that the GWAS data sets used contained a sufficiently clear polygenic signal contributing toward heritability. In the result section, it should show the details.

2. According to the tutorial of LDSC software, when estimating the genetic correlation between two diseases, all GWAS datasets should be ancestry-matched. As we know, the discovery stage of the schizophrenia PGC2 also contains samples from Japan, China and Singapore. This study only remove 5,582 control individuals were common to both datasets. I suggested they should remove the Asian samples in PGC2 and performed LDSC regression analysis again.

3. In the method section, it lacks the descriptions of these two GWAS of schizophrenia and ALS. For example, it should describe the number of the cases and controls, the ancestry of the samples, the number of SNPs used in the analysis, etc.

Reviewer #1 (Remarks to the Author):

McLaughlin and colleagues have performed a thorough study examining the genetic overlap between amyotrophic lateral sclerosis (ALS) and schizophrenia. They examined genome-wide association study (GWAS) datasets from 36,052 individuals in an ALS cohort and 79,845 individuals in a schizophrenia cohort. Using linkage disequilibrium (LD) score regression they demonstrated an overlap of 14.3% between ALS and schizophrenia, and with polygenic risk scores (PRS) they explained up to 0.12% of the variance in ALS. Their results seem to indicate that this overlap is specific to schizophrenia, since a similar overlap was not observed when comparing their cohort of ALS patients to cohorts of patients with Alzheimer's disease (AD), attention deficit-hyperactivity disorder (ADHD), autism spectrum disorder (ASD), bipolar disorder (BPD), major depressive disorder (MDD), or multiple sclerosis (MS). Additionally, they identified new potential loci that were shared between patients with ALS and schizophrenia (CNTN6, TNIP1, PPP2R2D, NCKAP5L, and ZNF295-AS1). The findings described by McLaughlin and colleagues are interesting, and this is a clear, concise, and well-written manuscript.

We thank the reviewer for their encouraging words and for the comprehensive review of our manuscript. We have responded to the individual concerns of the reviewer below and by editing the main manuscript and supplementary information.

Comments:

1 Abstract:

1a Even though findings can be significant, they do not have to be clinically meaningful. A genetic correlation between ALS and schizophrenia of 14.3% seems reasonable, but since the PRS only explains up to 0.12% of the variance in ALS, one could wonder whether this relatively small percentage is clinically meaningful. Maybe the authors could comment on the relevance of their findings?

We understand and respect the reviewer's viewpoint. While an explained variance of 0.12% has low predictive accuracy for ALS given schizophrenia PRS (and explained variance from PRS is inevitably always lower than h^2 or r_g estimates; see Wray et al [2013] Nat Rev Genet 14:507-515) we respectfully point out that disease prediction is not the purpose of the study. Rather, our findings make the important novel observation that, even within the weak signal of polygenic heritability in ALS (8.2%) and schizophrenia (23%), the underlying genetic aetiologies of both diseases are overlapping. This truly novel finding is of fundamental importance to our understanding of the pathophysiological mechanisms of both diseases, with respect to taxonomy and in patient sub-stratification in the design of precision medicine approaches towards new therapeutics. The latter consideration is of considerable importance both in ALS and schizophrenia, as they are increasingly recognised as clinical syndromes rather than individual disease entities. Our findings, when considered alongside previous case-control studies showing increased aggregation of neuropsychiatric conditions in ALS pedigrees, provide additional genomic strategies for segregation of subcohorts based on the presence of susceptibility to network disruption, manifesting as early life schizophrenia in some family members, and late onset neurodegeneration in others.

We also draw the reviewer's attention to a further important consideration, namely the limitation of using whole-genome SNP data. Such data, although the best currently available for large cohorts, are limited to the capture of information about genetic mechanisms of disease which are conferred by common genetic variation. We and others have shown that the "missing heritability" of both ALS and schizophrenia is substantial, and likely to be conferred by rare genetic variation, *de novo* variants, genetic variation that cannot be tagged by common SNPs and environmental risk, including gene-environment interaction. Accordingly, and based on our clinical and genomic findings, we suggest that many of the undiscovered genetic and environmental contributors to both diseases (and their underlying disease mechanisms) are also shared, and our findings therefore provide rationale for future work directed at identifying these aetiological agents.

1b Because only 0.12% of the variance in ALS appears to be explained by schizophrenia PRS, the statement that "shared neurobiological mechanisms between these two disorders will be relevant to future preclinical and clinical studies and in novel drug discovery" seems too strong. Please adjust accordingly.

On reflection, we agree that this statement is better made in a more general context. The sentence has been amended so that it now says, "It is likely that shared neurobiological mechanisms between these two disorders will engender novel hypotheses in future preclinical and clinical studies." (**Manuscript lines 9-11**)

2 Introduction:

2a Maybe the authors could elaborate on how they define the additive polygenic risk conferred by common genetic variants? Their definition is probably similar to the definition used in large-scale studies focusing on other diseases, but quite different from the definition generally used in the ALS field. To assist readers it would help to include a definition, especially since the authors are using summary data (at least for a subset of samples and analyses), and consequently, it is important for readers to realize that they are not referring to the presence of multiple variants within a single individual.

We thank the reviewer for this helpful suggestion. To improve the reader's overall understanding of the definition that we have used in the study, we have included a parenthetical statement explaining additive polygenic risk ("many risk-increasing alleles of low individual effect combining to cause disease") where it is mentioned in the second paragraph of the introduction. (**Manuscript line 24**)

2b In the last sentence of their introduction, the authors mention that they "demonstrate that this polygenic overlap with ALS is specific to schizophrenia". This statement seems too strong, especially since the number of subjects examined with other diseases might have been too small (limited power). Please see the Results section for more detailed comments.

We fully agree with the reviewer that the statement is too definitive. Our clinical data supports a polygenic overlap between ALS and other neuropsychiatric conditions, and we agree that larger studies in the future might yield genetic correlations between ALS and other neuropsychiatric traits. We have therefore restructured and revised the relevant sentence in the introduction to provide a more generalized statement that incorporates the possibility of limited power in the other secondary phenotypes. The sentence now reads: "We provide evidence for genetic correlation between the two disorders which is unlikely to be driven by diagnostic misclassification and we demonstrate a lack of polygenic overlap between ALS and other neuropsychiatric and neurological conditions, which could be due to limited power given the smaller cohort sizes for these studies." (**Manuscript lines 36-40**)

2c It might be useful to include a brief description of LD score regression upon its first introduction in the text. For instance, that LD score regression is used because it is able to distinguish confounding from polygenicity, which makes sense since "inflation from cryptic relatedness within or between cohort or population stratification purely

from genetic drift will not correlate with LD" (Bulik-Sullivan et al., Nature Genetics, 2015). Variants in LD with a causal variant will show an elevation in test statistics in association analysis proportional to their LD with the causal variant.

We thank the reviewer for these insightful suggestions. On the first mention of LD score regression early in the narrative of the Results section, we have extended its description to include the requested details. The full description now reads: "We first used linkage disequilibrium (LD) score regression with ALS and schizophrenia summary statistics; this technique models, for polygenic traits, a linear relationship between a SNP's LD score (the amount of genetic variation that it captures) and its GWAS test statistic. This distinguishes confounding from polygenicity in GWAS inflation and the regression coefficient can be used to estimate the SNP-based heritability (h_s^2) for single traits. In the bivariate case, the regression coefficient estimates genetic covariance (ρ_g) for pairs of traits, from which genetic correlation (r_g) is estimated; these estimates are unaffected by sample overlap between traits." (Manuscript lines 47-54)

3 Results:

3a The authors show that the schizophrenia-based PRS accounts for 0.12% of the phenotypic variance in the ALS patients. Are the authors able to test the opposite, determine the ALS-based PRS in schizophrenia patients?

We agree with the reviewer that this would be a useful study. However, access to individual-level genotype data for the schizophrenia cohort is strictly controlled by the Psychiatric Genomics Consortium (PGC) and our access for this study was expressly limited to the purpose of identifying and removing from the ALS dataset individuals that were either duplicated or genetically related in the two GWAS cohorts. Because of this, testing ALS-based PRS in the schizophrenia cohort was beyond the scope of the data access agreement for the current study. However, we are reassured by previous reports that have consistently shown that the profiles of variance explained by PRS between polygenic traits are relatively symmetrical, ie the variance explained in trait A given PRS for trait B is similar to the variance explained in trait B given PRS for trait A. An example of this (from *The Lancet* [2013] 381:1371-1379) is included below, showing this symmetry between pairs of five neuropsychiatric traits. Therefore we would expect a similar symmetry for ALS PRS in schizophrenia. Accordingly, we respectfully propose that while it would be an interesting addition to our study, the exclusion of ALS PRS in the schizophrenia cohort due to data access restrictions does not detract from the overall significance of our findings.

Figure 3: Pair-wise cross-disorder polygene analysis
 We derived polygene risk scores for each disorder (discovery sets) and applied them sequentially to the remaining disorders (target sets). Results are grouped by each discovery set. Each pair is shown on the x-axis and the proportion of variance explained for the target disorder (estimated via Nagelkerke's pseudo R²) on the y-axis. For purposes of illustration, three p₁ cutoffs are shown, but appendix p 62 shows the proportion of variance results for a broader range of cutoffs. p₁=training-set p value (used to select training set SNPs). Significance of results: a=p<0.05; b=p<10⁻⁶; c=p<10⁻⁸; d=p<10⁻¹²; e=p<10⁻¹⁶; f=p<10⁻³⁶. ADHD=attention deficit-hyperactivity disorder. ASD=autism spectrum disorders. BPD=bipolar disorder. MDD=major depressive disorder.

3b The authors included information about Breaking Up Heterogeneous Mixture Based On Cross-locus correlations (BUHMBOX) in the supplementary information, but it might be useful to include a concise description in the main text, including a definition for heterogeneity (e.g., one disease caused by different mutations) and for pleiotropy (e.g., one mutation causing different diseases).

We thank the reviewer for this suggestion. A brief description of the method is now included with the first mention of BUHMBOX in the third paragraph of the Results. The full sentence now reads: "Using BUHMBOX, a tool that distinguishes true genetic relationships between diseases (pleiotropy) from spurious relationships resulting from heterogeneous mixing of disease cohorts, we determined that misdiagnosed cases in the schizophrenia cohort (for example young-onset FTD-ALS) did not drive the genetic correlation estimate between ALS and schizophrenia ($p = 0.94$)."
(Manuscript lines 79-82)

3c They stress that if the genetic overlap between ALS and schizophrenia is driven by misdiagnosis of ALS as schizophrenia, then the percentage of misdiagnosis has to be around 5%. ALS and schizophrenia are clinically very different and 5% does seem unlikely; however, since a subset of ALS patients demonstrates signs of frontal temporal dementia (FTD; for example, behavioral variant FTD [bvFTD]), one cannot exclude the possibility that some of those cases who developed symptoms of FTD, possibly before they exhibited symptoms of ALS, could be misdiagnosed with schizophrenia. Although it remains unlikely that misdiagnosis explains the entire overlap (especially because of the difference in age at onset), it could, potentially, explain a proportion of the overlap.

We agree with the reviewer that it is important not to dismiss the possibility of misdiagnosis in the schizophrenia cohort, no matter how unlikely. Indeed, this is the very reason for which we explored this possibility using our data in the Discussion. From a clinical perspective, the possibility of misdiagnosis is very unlikely. Samples included in the ALS study were from patients fulfilling the El Escorial Criteria for ALS. The mean age of onset of ALS is 62, and those with atypical presentations, or those with behavioural variant FTD and limited evidence of motor involvement are generally excluded from our datasets. Similarly, the PGC dataset only includes typical cases of schizophrenia, the age of onset of which is in early adulthood, and those with neurological findings suggesting a system degeneration are excluded. However, we acknowledge the very remote possibility of misdiagnosis. We have added a clause to the end of the 5th paragraph in the Discussion (concerning the possibility of misdiagnosis), so that the final sentence now states: "We are therefore confident that this genetic correlation estimate reflects a genuine polygenic overlap between the two diseases and is not a feature of cohort ascertainment, but the possibility of some misdiagnosis in either cohort cannot be entirely excluded based on available data."
(Manuscript lines 149-150)

3d It is unclear why the authors did not put more emphasis on the potential novel loci. It would be interesting, for instance, to investigate whether there is an increased burden of rare variants in identified genes (e.g., CNTN6, TNIP1, PPP2R2D, NCKAP5L, and ZNF295-AS1). Moreover, please provide a brief explanation why a conditional false discovery rate (cFDR) of less than 0.01 was used.

While we stand behind the robustness of our cFDR results, we intend for the main message of the paper to be the overall polygenic genetic correlation between ALS and schizophrenia, which, as we demonstrate with LD score regression and polygenic risk scores, is derived mainly from the genome-wide signal of inflation in (weak) GWAS statistics. We have included cFDR analysis and resulting novel ALS-associated loci so that we may catalyse further research into these pleiotropic disease risk loci, but we would like to emphasize that much more needs to be understood about the mechanisms conferring pleiotropic risk of disease at individual loci before comprehensive statements about causality can be made. Nevertheless, we believe that inclusion of these data in this study provides an early indication of the importance and likely success of fine-grained investigation of the link between ALS and schizophrenia that we have evidenced with our genetic correlation estimate and PRS analyses.

We chose 0.01 as our cFDR threshold following Andreassen et al (the original study utilizing this method); this has now been stated in the final paragraph of the Methods section.

3e The authors write that "to investigate the polygenic overlap between ALS and schizophrenia, we used individual-level and summary data from GWAS for ALS ... and schizophrenia ..." Maybe the authors could make it easier for readers to understand for which analyses they used individual-level data and for which analyses summary data?

We have made three edits in the Results narrative to clarify the use of summary and individual-level data:

1. "We first used linkage disequilibrium (LD) score regression with ALS and schizophrenia summary statistics..." in second paragraph; (**Manuscript lines 47-48**)
2. "In addition to schizophrenia, we estimated genetic correlation with ALS using GWAS summary statistics..." in the third paragraph; (**Manuscript lines 65-66**)
3. "PRS calculated on schizophrenia GWAS summary statistics for twelve p-value thresholds (P_T) explained up to 0.12% ($P_T = 0.2$, $p = 8.4 \times 10^{-7}$) of the phenotypic variance in the individual-level ALS genotype data..." in the third paragraph. (**Manuscript lines 72-75**)

3f It is confusing that the authors refer to Supplementary Information 2, before they refer to Supplementary Information 1. Please put the Supplementary Information in order.

Thank you for pointing this out. This has been amended in the main text (changes tracked; **manuscript lines 46, 61, 188, 198, 201**) and in the supplementary information (not tracked). NB supplementary information 7 is called before supplementary information 3-6; this is because of the structure of the supplementary information – technical details come first, then information about funding and consortia. SI7 concerns funding/consortium information about the IGAP dataset, but requires referencing upon mention of the dataset in the Results section.

3g Based on their results, it does not seem fair to conclude that the genetic correlation between ALS and schizophrenia is specific to schizophrenia. Figure 1 demonstrates substantial error bars for most comparisons, which might be a reflection of the relatively small sample size of some studies (e.g., 5,422 individuals in the ADHD cohort). The effect of sample size is emphasized when examining the height variable: there is no reason to assume that height demonstrates a significant correlation with ALS, but because of the relatively large sample size (253,288) even a small difference almost reaches significance. In their Discussion, the authors do mention that the absence of an overlap "could relate to statistical power conferred by secondary phenotype cohort sizes", which is correct. They should, therefore, be more careful with the interpretation of their findings.

We agree with this concern (see also our response to concern 2b), and our statement about statistical power was made with the exact intention of encouraging caution in interpretation. In our revised manuscript, there is now no use of the word "specific" and the fourth paragraph of the Discussion is entirely structured around the very argument that concerns the reviewer.

We trust that the reviewer agrees that the manuscript now appropriately addresses the possibility of limited power in the other secondary cohorts, and that future studies may unveil a broader genetic relationship between ALS and neuropsychiatric traits.

4 Discussion:

4a The Discussion contains statements without references. Please add additional references.

An additional eight references have been added to support the statements made in the Discussion.

4b As mentioned previously, it is true that it seems unlikely that misdiagnosis explains the majority of the overlap, but the authors cannot exclude the possibility that misdiagnosis might account for a (small) proportion of the overlap.

Agreed. Please see our response to comment 3c.

5 Methods (including Supplementary Information):

5a McLaughlin and colleagues constrained the intercept of their mixed linear model (MLM), because they wanted to be conservative. It has been reported, however, that "constrained intercept LD score regression has lower standard error - often by as much as 30% - than LD score regression with unconstrained intercept, but will yield biased and misleading estimates if the intercept is misspecified, e.g., if we ... do not completely control for population stratification" (Bulik-Sullivan et al., Nature Genetics, 2015). Maybe the authors could comment on this and explain how their results would change if they would not use a constrained model?

In order that the reader may make an informed decision on the validity of our approach in the context of this important question, the supplementary information contains a complete report of all heritability, genetic covariance and genetic correlation estimates for both constrained and free-intercept methods with meta-analysis and mixed linear model summary statistics (supplementary tables 1 and 2). The difference that a non-constrained intercept makes is that the SNP-based heritability estimate for ALS is lower (owing to its rare variant architecture), which results in a **higher** genetic correlation estimate ($r_g = \frac{\rho_g}{\sqrt{h_A^2 h_B^2}}$) with schizophrenia, which we consider to be unrepresentative of the true scenario given that our constrained model most accurately replicates REML-derived SNP-based heritability estimates for ALS (van Rheenen et al. [2016] Nat Genet 48:1043-1048). We therefore feel that the most prudent approach is to report as our main finding the lower estimate (based on constrained-intercept h^2 estimation), but to provide, in addition, a complete breakdown of all estimates based on alternative models for the discerning reader. These numbers are provided in supplementary tables 1 and 2.

On the subject of population stratification, in addition to using mixed linear model association statistics and principal components when appropriate, we have further dealt with this possibility in two ways.

1. With both LD score regression and PRS, we have repeated analyses using permuted data. With LD score regression, the use of summary statistics calculated in randomly permuted cases and controls resulted in null SNP-based heritability estimates (supplementary table 1); therefore, genetic correlation based on this SNP-based heritability cannot exist. This suggests that our findings are a consequence of genuine case-control mediated inflation in the GWAS summary statistics, and not population stratification. With PRS, all estimates of explained variance were non-significant and close to zero when schizophrenia PRS was assessed in randomly permuted case-control data (supplementary figure 3), indicating that our PRS findings also reflect genuine signals derived from disease status and not population stratification.
2. We also ruled out population stratification with a new analysis, now included as supplementary figure 1 (see also response to comment 2 from reviewer #2). Here, we repeated our genetic correlation analysis using an older dataset comprising 21,856 individuals of exclusively European ancestry (Ripke et al. [2011] Nat Genet 43(10):969-976) instead of the larger cohort of 79,845 individuals of European and Asian ancestry. When we use this dataset to estimate genetic correlation the results are very similar, indicating that the inclusion of Asian-ancestry individuals in the main study cohort did not influence the outcome of our analysis.

5b Since the authors also identified five loci already known to be involved in ALS, including C9ORF72 and TANK binding kinase 1 (TBK1), it seems that they have not excluded patients with known mutations, nor have they removed (other) loci known to be involved in ALS. It would be interesting to perform a sensitivity analysis to

determine what happens after exclusion of known loci. How would this sensitivity analysis influence the SNP-based heritability, the genetic correlation, and PRS? This is particularly relevant because of reports that have described patients with a repeat expansion in *C9ORF72* with atypical symptoms, such as schizophrenia and psychotic illnesses. It would also ensure that the genetic correlation is not driven by a single locus (or a few loci) or biased by the fit of the model (Hagenaars et al., *Molecular Psychiatry*, 2016). Have the authors considered removing all genome-wide significant loci?

We absolutely agree that monogenic overlap between ALS and schizophrenia (such as the *C9orf72* positive reports of psychosis that the reviewer describes) are an important and interesting consideration in the context of our findings of polygenic overlap between the two diseases. However, given that ALS GWAS has (to date) identified very few genome-wide significant loci (five in total), we do not expect these to significantly influence our results, which are primarily driven by the polygenic signal from potentially thousands of weakly-associated loci.

To demonstrate this concretely, we have followed the reviewer's suggestion and repeated our LD score regression analysis after removal of all SNPs in LD ($r^2 > 0.2$) with the five genome-wide significant loci reported in van Rheenen et al. [2016] *Nat Genet* 48:1043-1048. We find that the genetic correlation estimates barely change for all models and summary statistics, reaffirming the principle that the signal is driven by many weakly-associated loci spread across the genome. This is illustrated in the figure below.

Although we could report these observations in the supplementary material if required, it would be our preference to omit these results, as we feel they might distract the reader from the overall message for three reasons:

1. The polygenic signal that our genetic correlation estimates describe might involve some of the (weaker) genome-wide significant loci. For example, the newly-discovered *MOBP* locus, which is one of just five genome-wide significant loci in the recent 2016 ALS GWAS, may in fact simply be one of the first of many polygenic risk loci to be discovered by increasingly large GWAS.
2. It is unconventional in studies of this kind (polygenic genetic correlation studies) to remove loci that exceed significance thresholds. Larger GWAS, such as the 2014 schizophrenia GWAS that identified 108 genome-wide significant loci, consider these loci to be part of the overall polygenic architecture of the disease. This may be the case for some of the "significant" ALS loci.
3. LD score regression actually already takes some steps to avoid over-counting highly significant loci. This is through the use of regression weights to down-weight high-LD SNPs (often also highly significant in GWAS statistics) and by hard-filtering highly significant loci (usually $\chi^2 > 80$ or ~ 30 if using the in-built two-step estimator). The potential problem of highly significant loci is therefore somewhat ameliorated by using this technique.

5c It looks as if the supplementary data is formatted for another journal, please use "." instead of "·".

Thank you for pointing this out. This has been amended (changes not tracked).

5d It is unclear whether subjects excluded from the ALS GWAS from the Finnish and German strata were present in the schizophrenia group or in the control group of the schizophrenia cohort. Why were those subjects not excluded from the schizophrenia dataset instead of from the ALS dataset, since the ALS dataset is smaller? Due to the lack of individual-level data for a subset of samples?

This is indeed the reason. The Psychiatric Genomics Consortium (PGC) permitted access to the schizophrenia dataset so that we could identify and remove individuals from the ALS data that were either related to PGC subjects or duplicated (in the case of shared control cohorts). This data access permission did not extend to the German and Finnish strata within the PGC data, so relatedness or duplicates could not be identified (but some overlap was suspected or could not be ruled out). We therefore, instead, removed all Finnish and German individuals from the ALS dataset to mitigate the risk of spurious results in our PRS analysis.

5e The Supplementary Information is very helpful and complete, and the legends are brief, but informative. Sometimes, however, the order of the rows in tables varies, which can be confusing. For instance, Supplementary Tables 1 and 2 start with MLM constrained, but Supplementary Table 7 starts with MLM free.

We thank the reviewer for drawing our attention to this error. The rows of supplementary tables are now all in the same order (changes not tracked).

6 Typos:

6a Occasionally, references are not inserted properly, e.g., "cohort-level logistic regression¹²." instead of "cohort-level logistic regression.¹²"

Thank you to the reviewer for pointing this out. This has been amended where relevant in the manuscript.

6b The authors already use "SNP" in their Introduction, but the abbreviation is introduced in the Results section. Please introduce all abbreviations upon their first occurrence in the text.

We have amended the text so that the term "SNP" is defined in its first usage. (**Manuscript line 25**)

Reviewer #2 (Remarks to the Author):

1 This study did not show the details of estimated heritability calculated by using the LDSC. A heritability Z score of > 0.4 and a mean χ^2 statistic of >1.02 were usually used to ensure that the GWAS data sets used contained a sufficiently clear polygenic signal contributing toward heritability. In the result section, it should show the details.

We thank the reviewer for taking the time to review our manuscript. We have now included the mean χ^2 statistic (1.1.3) along with the heritability results (**Manuscript line 56**) so that the reader has complete information on the extent of the polygenic signal. The 95% confidence interval is reported for the heritability estimate (7.2-9.1%), which corresponds to a heritability Z-score of 16.1. As all other statistics report 95% confidence intervals, we respectfully suggest that, for consistency, these estimates are reported in the place of a Z-score (the two values represent the same properties of the estimate).

2 According to the tutorial of LDSC software, when estimating the genetic correlation between two diseases, all GWAS datasets should be ancestry-matched. As we know, the discovery stage of the schizophrenia PGC2 also contains samples from Japan, China and Singapore. This study only remove 5,582 control individuals were common to both datasets. I suggested they should remove the Asian samples in PGC2 and performed LDSC regression analysis again.

While we recognize this limitation, we were unable to identify/remove individuals of Asian ancestry from the schizophrenia cohort due to data access restrictions with the Psychiatric Genomics Consortium (please see response to comment 3a from Reviewer #1). However, the availability of the 2011 “PGC1” dataset (Ripke et al. [2011] Nat Genet 43(10):969-976) has allowed us to perform an additional analysis of 21,856 individuals of exclusively European ancestry. We find our genetic correlation estimate to be very similar to our 2014 “PGC2” estimate, which indicates that the inclusion of individuals of Asian ancestry do not significantly bias the results.

We agree with the reviewer that this is an important consideration so we have now included this finding in our study and the supplementary information, with the following sentence in paragraph 3 of the Results section: “Results were similar for a smaller schizophrenia cohort of European ancestry (21,856 individuals), indicating that the inclusion of individuals of Asian ancestry in the schizophrenia cohort did not bias this result (supplementary figure 1).” (**Manuscript lines 63-65**) For the reviewer’s convenience, a copy of this figure is shown below.

Supplementary figure 1 Comparison of genetic correlation estimates for two schizophrenia cohorts. SCZ1 denotes data from the 2011 study by the Schizophrenia Psychiatric Genome-Wide Association Study Consortium (European ancestry cohort); SCZ2 indicates data from the 2014 study by the Schizophrenia Working Group of the Psychiatric Genomics Consortium (European and Asian ancestry cohort). Error bars indicate 95% confidence intervals.

3 In the method section, it lacks the descriptions of these two GWAS of schizophrenia and ALS. For example, it should describe the number of the cases and controls, the ancestry of the samples, the number of SNPs used in the analysis, etc.

The first paragraph of the Methods section (Study population and genetic data) has been updated to include these details. (**Manuscript lines 177-180**)

Reviewers' Comments:

Reviewer #1 (Remarks to the Author)

McLaughlin and colleagues addressed most comments properly. In their response to referees letter, they clearly explained why certain decisions were made, they provided helpful information, and when necessary, they changed their text accordingly; for instance, their statements are currently more conservative than in the previous version of their manuscript. Although it would have been great if they could have put more emphasis on the potential novel loci (e.g., by determining the burden of rare variants), their revised manuscript, which highlights a new genetic overlap between amyotrophic lateral sclerosis (ALS) and schizophrenia, seems suitable for publication.

Reviewer #2 (Remarks to the Author)

In my opinion, I agree most of the authors' responses to reviewers' comments. However, the authors should also further discuss the limitation of the implications of the relative LD score or PRS, which just explain less genetic variety or overlap of ALS and schizophrenia. Especially when we consider the heterogeneity across populations, using the open PGC data, the clinical implication should also be interpreted for general readers.

The manuscript can be principally accepted after further polishing.

Reviewer #1 (Remarks to the Author):

McLaughlin and colleagues addressed most comments properly. In their response to referees letter, they clearly explained why certain decisions were made, they provided helpful information, and when necessary, they changed their text accordingly; for instance, their statements are currently more conservative than in the previous version of their manuscript. Although it would have been great if they could have put more emphasis on the potential novel loci (e.g., by determining the burden of rare variants), their revised manuscript, which highlights a new genetic overlap between amyotrophic lateral sclerosis (ALS) and schizophrenia, seems suitable for publication.

We thank the reviewer for taking the time to examine our manuscript a second time. We are glad that we have satisfied their concerns and we agree that the statements made in the manuscript are better made more conservatively. We agree that burden analysis in these novel loci would be very interesting and important follow-up work, but to do this would require access to large-scale whole-exome or whole-genome sequencing data instead of GWAS data. Projects are currently underway to generate such data, but the timescales are prohibitively long (several more years) to realistically incorporate any such analysis into the current study. Nevertheless, we have highlighted the importance of this observation made by the reviewer by adding a clause in the Discussion, such that the end of the third paragraph (lines 134-138) now reads: "Further investigation into the biological roles of these genes may yield novel insight into the pathophysiology of certain subtypes of ALS and schizophrenia, and as whole-genome and exome data sets become available in the future for appropriately large ALS case-control cohorts, testing for burden of rare genetic variation across these genes will be particularly instructive, especially given the role that rare variants appear to play in the pathophysiology of ALS."

Reviewer #2 (Remarks to the Author):

In my opinion, I agree most of the authors' responses to reviewers' comments. However, the authors should also further discuss the limitation of the implications of the relative LD score or PRS, which just explain less genetic variety or overlap of ALS and schizophrenia. Especially when we consider the heterogeneity across populations, using the open PGC data, the clinical implication should also be interpreted for general readers.

The manuscript can be principally accepted after further polishing.

We thank the reviewer for their additional comments and for taking the time to read our revised manuscript. In response to these final suggestions, we have added a sentence to the first paragraph of the Discussion (lines 105-111) to explain the implication of the difference between our LD score regression-based genetic correlation estimate and our PRS-based explained variance. This addition also further discusses the clinical implications of the findings. This section of the paragraph now reads: "Given that our genetic correlation estimate relates to the polygenic components of ALS ($h_S^2 = 8.2\%$) and schizophrenia ($h_S^2 = 23\%$) and these estimates do not represent all heritability for both diseases, the accuracy of using schizophrenia-based PRS to predict ALS status in any patient is expected to be low (Nagelkerke's $R^2 = 0.12\%$ for $P_T = 0.2$), although statistically significant ($p = 8.4 \times 10^{-7}$). Nevertheless, the positive genetic correlation of 14.3% indicates that the direction of effect of risk-increasing and protective alleles is consistently aligned between ALS and schizophrenia, suggesting convergent biological mechanisms between the two diseases."